# Configuration analysis of crop-pollination service management: a novel insight from the theory of planned behavior

Hongkun Zhao[1], Yaofeng Yang[2], Yajuan Chen [1,2]*, Huyang Yu[1], Zhuo Chen[1], Zhenwei Yang[3]

**1** School of Economics and Management, Inner Mongolia Normal University, Hohhot, China, **2** Northwest Institute of Historical Environment and Socio-Economic Development, Shaanxi Normal University, Xi'an, China, **3** College of Computer Science and Technology, Inner Mongolia University for Nationalities, Tongliao, China

☯ These authors equally contributed to this work
* yaya576@126.com

**Data availability statement:** All relevant data are within the paper and its Supporting Information files.

**Funding:** Yajuan Chen: National Natural Science Foundation of China (No. 32060317), the Inner Mongolia Natural Science Foundation

## Abstract

As the crisis of crop-pollination service increasingly gains global attention, improving crop-pollination service management (CPSM) has become a key challenge to achieve sustainable agriculture and safeguard food supply. Given that farmers are directly responsible for making decisions and managing agriculture, strategies for promoting CPSM should consider their perceptions, knowledge and role in enhancing pollination. A survey of 267 randomly selected smallholder farmers in Dengkou County was conducted to create and evaluate an integrated index for assessing on-farm pollination management among farmers, and to explore how key factors, grounded in the extending the theory of planned behavior (TPB), can influence their CPSM behaviors. The data is analyzed by using regression analysis, necessary condition analysis, fuzzy-set qualitative comparative analysis (NCA-fsQCA), and independent sample T test, and the findings reveal that education level and agricultural acreage are positively correlated with CPSM; there are three causal configurations to enhance CPSM: AT & PBC path, AT & Economic Incentive path, and PBC & Economic Incentive path; the contrasting effects of antecedent variables on different groups of principles of CPSM; the optimal state of CPSM requires at least Economic Incentive $1900.27. The findings provide practical implications for enhancing CPSM among different farmers through multi-pathways. This study can help to formulate CPSM strategies and increase farmers' participation in pollinator-supporting behaviors in actual agricultural cultivation.

(No. 2023MS04014), and The Fundamental Research Funds for the Inner Mongolia Normal University (No. 32150022210).

## 1. Introduction

The United Nation's Millennium Ecosystem Assessment program greatly stimulated ecosystem services research and firmly reminded that without ecosystem services there would be no human life [1]. Ecosystem services related to agriculture, especially crop-pollination service have particularly significant implications in sustaining food production. In context of high human demands for different crops due to population growth and changing dietary preferences, about 35% of global crops that provide nutrients rely on insect-mediated pollination [2–4]. Meanwhile, intensification and expansion areas of agriculture have increased harvest, but also have emerged as major causes of biodiversity loss among pollinating insects and have adversely affected the yield stability of insect-pollinated crops [5]. Additionally, crops that depend on pollinators exhibit higher rates of agricultural expansion compared to non-pollinator-dependent crops [6]. It means that enhancement of crop-pollination service has become a key challenge to achieving sustainable agriculture and safeguarding food supply. As farmers are direct beneficiaries and ultimate managers of agriculture at local scale, it is essential to understand farmers' perceptions, knowledge, and practices of crop-pollination service and explore factors behind them [7]. Therefore, sustainable agriculture should incorporate more nuanced and comprehensive understandings of farmers' role in enhancing crop-pollination service, highlighting the need to increase engagement and trust of farmers in implementing pollinator-supporting practices.

Crop-pollination is a vital ecosystem service required by many crops, particularly those that rely on insect pollination, such as sunflowers and melons. Historically, natural crop-pollination by wild pollinators such as bees, butterflies and flies has played a central role in maintaining agricultural productivity and biodiversity [8]. This method of pollination, rooted in local ecosystems, is sustainable but vulnerable to threats such as habitat loss, pesticide use and climate change, which can reduce pollinator diversity and stability [9]. With the advent of industrial agriculture, crop-pollination has become increasingly dependent on human intervention, particularly through the use of managed pollinators such as honey bees [10]. While this shift has enabled higher yields and greater efficiency, it has also led to reduced resilience as monoculture practices and pesticide use undermine pollinator diversity. In addition, over-reliance on a few pollinator species in industrial agriculture limits the overall stability of pollination systems [11]. In response, artificial insect pollination techniques, such as the use of commercially reared pollinators or mechanical pollination, have been introduced in areas where natural pollinators are insufficient [12]. However, these methods are often more expensive and may not provide the same ecological benefits as wild pollination. Balancing these systems while protecting and enhancing natural pollination services will be key to meeting growing environmental challenges and increasing agricultural demands.

To counteract the decrease in crop-pollination service, researchers and policymakers have prioritized three main strategies of crop-pollination service management (CPSM; The abbreviations in this paper are shown in Table 1): the management of pollinator habitats, which provides supplementary resources (such as feeding and

**Table 1. List of abbreviations.**

| CPSM | Crop-pollination service management |
|------|-------------------------------------|
| NCA | Necessary condition analysis |
| fsQCA | Qualitative Comparative Analysis of Fuzzy Sets |
| TPB | Theory of planned behavior |
| AT | Attitudes |
| SN | Subjective norm |
| PBC | Perceived behavioral control |

nesting resources) to pollinators for spatially aggregating [13], the management of pollinator species, which is based on relationships between crop species and pollinator community characteristics [14,15], and the management of jointness pollinator with other aspects, which is based on synergistic interactions between crop-pollination and soil factors or pest control [16]. From farmers' perspectives, whatever kind of management strategies should be represented by choices of farmers to maintain or introduce extensive pollinator-supporting practices. In general, more pollinator-supporting practices are supported largely through incentives, such as the form of agri-environment schemes, especially in North America and Europe. Nevertheless, farmers could be reluctant to implement these practices, even if they are proven to work well in improving crop-pollination service. While differences in perceptions between farmers and researchers or policymakers highlight an understanding and communication gap [17], these disparities also reflect variations in knowledge and back-ground among local farmers. This is because the benefits of pollinator-supporting practices on yield can vary depending on management factors, and farmers' behavior—an essential component of agricultural practices—is directly influenced by their local knowledge and experience [17]. Therefore, a general framework of CPSM with greater flexibility for farm-ers in pollinator-supporting behaviors should be developed, accounting for the trust (related to enhancing confidence in pollinator-supporting behaviors of farmers through understanding of pollination knowledge) and engagements (related to enhancing the trust to actively involve farmers in pollinator-supporting practices).

Additionally, what proposes a framework of CPSM is also a desire to promote a novel method for quantifying the farmers' on-farm pollination management to implement multiple pollinator-supporting practices and enhance cooperation between them on landscape scale. Since the framework of CPSM is a comprehensive framework with a view of multi-dimensional realities, measuring farmers' CPSM based on the framework will be important and challenging [16]. Previous studies have employed two methods for evaluating farmers' CPSM: directly asking farmers about their pollinator protection strategies or establishing indicators to assess the level of farmers' CPSM [18–20]. However, the two methods have limitations, maybe because of just provision a binary discrete value that doesn't capture the complexity of farmers' CPSM or because of simply utilized a weighted practice approach that doesn't fully address the complexity of interaction between farmers' perceptions, knowledge, and practices in CPSM. Therefore, to better capture complexity of CPSM, it is needed to develop an integrated index of CPSM based on the framework of CPSM for further investigating farmers' role in enhancing crop-pollination service.

As with most conservation problems, the challenge of enhancing crop-pollination service is fundamentally one of human behavior [21,22], so investigating farmers' role of CPSM requires interdisciplinary approaches [23,24]. Currently, social research around CPSM is in its infancy [25], and a few studies mainly focus on correlations between CPSM and demographics to elucidate its influencing factors. While these variables simplify the representation of potential factors in specific contexts, there is a lack of compelling evidence to explain the underlying mechanisms influencing farmers' CPSM due to the limited explanatory power of certain indicators [26]. In this context, the social psychology-based research paradigms to explore farmers' role of CPSM provides a holistic and transdisciplinary perspective, such as the theory of planned behavior (TPB). With greater behavioral predictability and fewer detection components than other methods, TPB has become one of the most widely used theories for analyzing farmers' behavior [27]. Meanwhile, when TPB is applied in specific case studies, some factors that can stimulate behavior change should be added to TPB to prevent

the solidification of theoretical framework [28]. As instrumental values of pollinators (economic benefits arising from crop pollination) are widely appreciated [13], it is essential to add variable (economic benefits) in TPB to improve interpretative ability in farmers' role of CPSM. Moreover, given limited society capacity of farmers to manage communal resource productivity, it is important for policymakers to identify the best subset of policies that can maximize farmers' CPSM levels [29,30]. At present, many studies use causal models (e.g., regression analysis and structural equation model) to examine relationships between variables, which is of concern the net effects between variables [31,32]. Actually, variance change of a variable is determined by the non-additive interaction of multi-factors, so net effect studies based on causal models ignore the complexity of factor action. Specifically, in CPSM research, central question is not which variable has the greatest net effect but how multi-combinations of conditions can help farmers to adapt and refine CPSM resulting in enhancement of crop-pollination service. Therefore, the fuzzy set quantitative comparative analysis (fsQCA) by using a configurational approach provides a deeper understanding of relationships between variables to better examine the causes of how to shape different CPSM scenarios [33,34]. Therefore, fsQCA is used to systematically examines the causes of different CPSM and the interactions between drivers thereby deepening understanding of configuration analysis of CPSM [33]. Meanwhile, necessary condition analysis (NCA) is used to complement fsQCA aiming to explore the extent to which single factors are necessary for farmers to render different CPSM scenarios [35].

Recognizing the looming pollination crisis around the world, a handful of governments in developed countries have grappled with pollinator conservation and CPSM issues. For example, the European Union has proposed a series of management practices to promote pollinator conservation and enhance crop-pollination services [36,37]. In practice, agriculture is a complex sector, with many different actors (farmers). Thus, it is necessary to incorporate farmers into the action (such as CPSM) which allows farmers to have greater flexibility in pollinator-supporting behaviors based on their knowledge, experience and perceptions about crop-pollination. Simultaneously, there is a need for further discussion to explore how farmers' socio-psychological factors influence their actions. Despite ecological research on the topic of pollinator conservation and pollination service management is abundant, there is few interdisciplinary research [22]. This study will fill the gap and its one objective is: (1) to propose an integrated index of CPSM based on the framework of CPSM, which allows to evaluate the state of CPSM which is implemented by farmers. Another objective of this study is: (2) to investigate and identify configurations of farmers' socio-psychological factors based on the TBP, which could better explain the state of CPSM. Specifically, it seeks to answer how farmers' socio-psychological factors could be configured to improve farmers' CPSM. This study will support refine CPSM strategies and enhance farmers' participation in pollinator conservation in real-world agricultural operations.

## 2. Theoretical background and hypotheses

### 2.1 A general framework of crop-pollination service management

Here, the framework of CPSM is provided, which allows farmers to have greater flexibility in pollinator-supporting behaviors to adapt specific landscape conditions, crop varieties, and management strategies. The framework of CPSM, which outlines how farmers trust in the beneficial effects of pollinator-supporting behaviors based on pollination knowledge, linked with their involvement in pollinator-supporting practices, is presented to promote pollinator conservation and enhance crop-pollination service. The four underpinning principles of the framework are respectively dependence, contribution, sensitivity, and execution (Fig 1). To better understand the framework of CPSM, the concepts of principles and the hypothesized links among them are explored below. Dependence represents farmers' cognition of the pollination dependency of their crops (Fig 1 Principle I). The pollinator dependence of different crops differs greatly in extent which range from little or no dependence (e.g., wind-pollinated or self-pollinated cereals) to partially dependence (e.g., melon, tomato, and sunflower) and are intrinsically linked to plant breeding systems [38,39]. Contribution refers to farmers' observations of the frequency of pollinator visits to crops (Fig 1 Principle II). The frequency of occurrence for pollinator's visits to crops is often context-dependent on specific landscape conditions and management strategies. Farmers' observations on the contribution

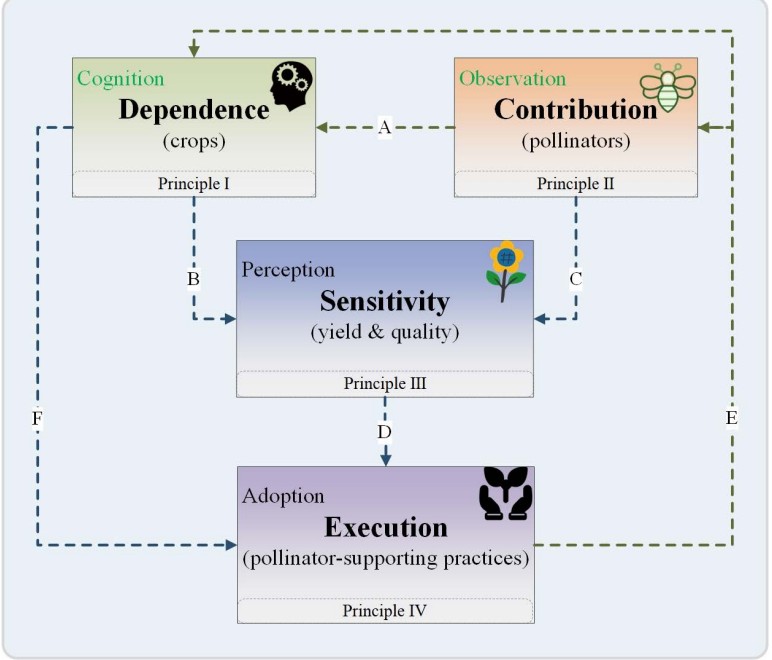

**Fig 1. A summarized general framework of CPSM.** The boxes: four underpinning principles of the framework of CPSM; A-F arrows: hypothesized links among four principles (the direction of blue arrows represents increasing farmers' trust in pollinator-supporting behaviors based on their pollination knowledge and experience; the direction of green arrows represents farmers' involvement in pollinator-supporting practices to enrich their related knowledge and experience). For further explanation of principles and links see text.

of pollinator-supporting behaviors are often essential to enhance farmers' knowledge and insights about dependence (Fig 1 arrow A). Sensitivity stands for farmers' perceptions of the extent of yield or quality reduction in absence of pollinators compared to potential yield or quality (Fig 1 Principle III). In general, a higher dependency on crop-pollination service could improve farmers' understanding of sensitivity in CPSM (Fig 1 arrow B). Indeed, sensitivity might also be relevant to potential effects of contribution changes on farmers' CPSM, and substituted technologies and other means (such as hand pollination) that farmers implemented in their fields (Fig 1 arrow C). Execution indicates farmers' adoption of agricultural practices that promote pollinator conservation and enhance crop-pollination service (pollinator-supporting practices) (Fig 1 Principle IV). Farmers' cognition of sensitivity is important for execution because it may affect individuals' perception of the effectiveness of adaptive actions, allowing farmers to take diverse pollinator-supporting practices in different CPSM scenarios (Fig 1 arrow D). The more diversified farmers' adoption of pollinator-supporting practices, the greater their options for avoiding adverse effects caused by insufficient pollinator contributions. This approach also helps to optimize the discrepancies between farmers' perceptions of pollinator dependence and those of researchers (Fig 1 arrow E). At the same time, farmers' willingness to adopt pollinator-supporting practices also is often limited by their knowledge of dependence (Fig 1 arrow F). Here, the framework of CPSM highlights a positive, and often neglected, feedback loop of the relationship between farmers' knowledge and experience and their pollinator-supporting behaviors (Fig 1 the direction of blue and green arrows). The four principles (dependence, contribution, sensitivity, and execution) within the positive feedback loop of the framework of CPSM may reinforce each other, offering insights into how to implement on-farm CPSM that are more realistic and more suited to local agriculture landscapes. Because the framework of CPSM is built to explore how farmers manage crop-pollination service, it is necessary to assess on-farm CPSM among farmers by using an integrated index based on the framework of CPSM. The algorithm for the index is presented in Section 3.3.

## 2.2 Theory of planned behavior and of the complexity theory

The TPB, a typical socio-psychological construct, was applied as the basic theoretical framework in analyzing farmers' CPSM. The TPB was proposed by Ajzen and Fishbein on the basis of the Theory of Reasoned Behavior (TRB) [40]. Five components are included in the TPB: attitude, subjective norm, perceived behavioral control, behavioral intention and actual behavior. Attitude is derived from behavioral beliefs (a positive or negative evaluation of the outcome), Subjective norm originates from normative beliefs (social forces from others or social institutions), and perceived behavioral control is closely related to control beliefs (the internal factors that promote or hinder behavior) [41]. Behavioral intention is defined as the anticipation of certain actions, which is collectively determined by the attitude, subjective norm and perceived behavioral control. According to the TPB, behavioral intention is the most critical predictor of actual behavior (Quine and Rubin, 1997). In addition, there may be interaction between attitude, subjective norm and perceived behavioral control [42]. Meanwhile, the three components can be also used to predict actual behavior together with behavioral intention [43]. TPB is highly relevant to CPSM as it helps explain how farmers' perceptions and behaviors regarding pollinator conservation are influenced by their attitudes, subjective norms, and perceived behavioral control. In the context of CPSM, the TPB framework enables the identification of key factors that shape farmers' intentions to adopt pollinator-supporting practices. Understanding these influences can help design more effective strategies to promote pollinator conservation and improve crop-pollination services at the farm level.

Previous studies have demonstrated that incorporating external incentives into TPB can better reveal special behavior [44,45]. However, the selection of indicators to extend TPB in a specific study largely depends on the characteristics of subject and goals. Dengkou County, located in the Inner Mongolia Autonomous Region of China, is an agricultural area known for its production of crops such as sunflowers, melons, and wheat. Since there is no governmental policy targeting crop-pollination in Dengkou County, economic benefit remains the primary pursuit of local farmers when making decisions. And economic efficiency plays a crucial role in implementing policies related to CPSM [46]. In fact, as an external variable, economic factors have been used to optimize the TPB framework [47]. Therefore, this study adds EI to TPB to provide a new perspective for extended TPB.

AT is defined as individuals' subjective evaluations of performing specific behaviors, which is a crucial variable in predicting behaviors [40,48]. In case of agricultural management, farmers' adoption of agricultural practices will increase if they have positive AT toward action [49]. Since the farmers' behavior can be significantly affected and identified by their attitude, H1 is proposed:

H1: Farmers' attitude (AT) has a positive relationship with their CPSM behavior.

SN represents individuals' perceptions of pressures or expectations from external groups and affairs [50]. These pressures or expectations arise from psychological conflict to perform target behaviors or not. Typically, SN resulting from positive external influences enhances individual behavior accordingly [51]. However, their intention to perform behaviors may decrease if behavior lacking policy guidance or contrary to social conventions [52,53]. SN not only directly reduces individual behaviors, but also reduces positive effects of AT and self-efficacy on actual behaviors [54]. So, H2 is proposed:

H2: The subjective norm (SN) from society has a negative relationship with farmers' degree of the implementation of CPSM.

PBC refers to the perception of one's ability to control target behaviors [41]. The stronger individuals' perceptions of control over target behaviors, the more behavioral intentions will arise [55]. In scenarios of agricultural practice, PBC accounts for the largest proportion of actual behavioral variance [56]. Thus, as farmers' beliefs about their control over CPSM increase, their related actions will also increase. So, H3 is proposed:

H3: Farmer's perceived behavioral control (PBC) has a positive relationship with the extent to which they implement CPSM.

EI reflects the influence of economic agents to engage in specific behaviors due to monetarily rewarded [57]. Economic Incentive provides individuals with the money needed for the behavior, which directly motivates the individual to enhance

that behavior [58]. As an agricultural operating practice characterized by improving ecological environment, the higher economic efficiency of CPSM, the higher farmers' level of CPSM [59]. So, H4 is proposed:

H4: Higher EI positively supports higher farmers' CPSM levels.

Complexity theory assumes that small random events (through positive feedback effects of increasing returns) can dynamically lead to multi-equilibria [60,61]. That is, complexity theory indicates that relationships between variables can be non-linear, and variables can produce different results in specific circumstances (for example, combinations of farmers' socio-psychological factors of different intensities may have impacts on their CPSM decisions leading to many forms of management decisions) [62,63]. Specifically, management elements vary among subjects (farmers), and it is up to the subjects to explore, react and continuously change their actions and strategies in response to different outcome [30]. Since agricultural management is a complicated socio-ecological system, enhancing and maintaining CPSM will require identifying multidimensional solutions instead of a single solution. This supports the formulation of differentiated schemes to enhance CPSM levels of different farmers based on different crop planting backgrounds and differentiated farmer characteristics. Therefore, different factors may form various CPSM solutions through complex combinations (Fig 2). Consequently, H5 and H6 are proposed:

H5: CPSM can be influenced by mutually complementary or alternative effects consisting of AT, SN, PBC, and EI.

H6: Better CPSM may or may not exist under same conditions across different configurations, depending on the specific manner in which farmers' Attitude (AT), Social Norms (SN), Perceived Behavioral Control (PBC), and Economic Incentive are integrated and balanced.

## 3. Data and methods

### 3.1 Study region

The study is conducted in Dengkou County, which located in the southwest part of Inner Mongolia Autonomous Region, China (Fig 3). The general situation is high in the southeast and low in the northwest, with an elevation of 1050m in the southeast and dropping to 1030m in the northwest. It is situated in the mid-latitude inland area, spanning

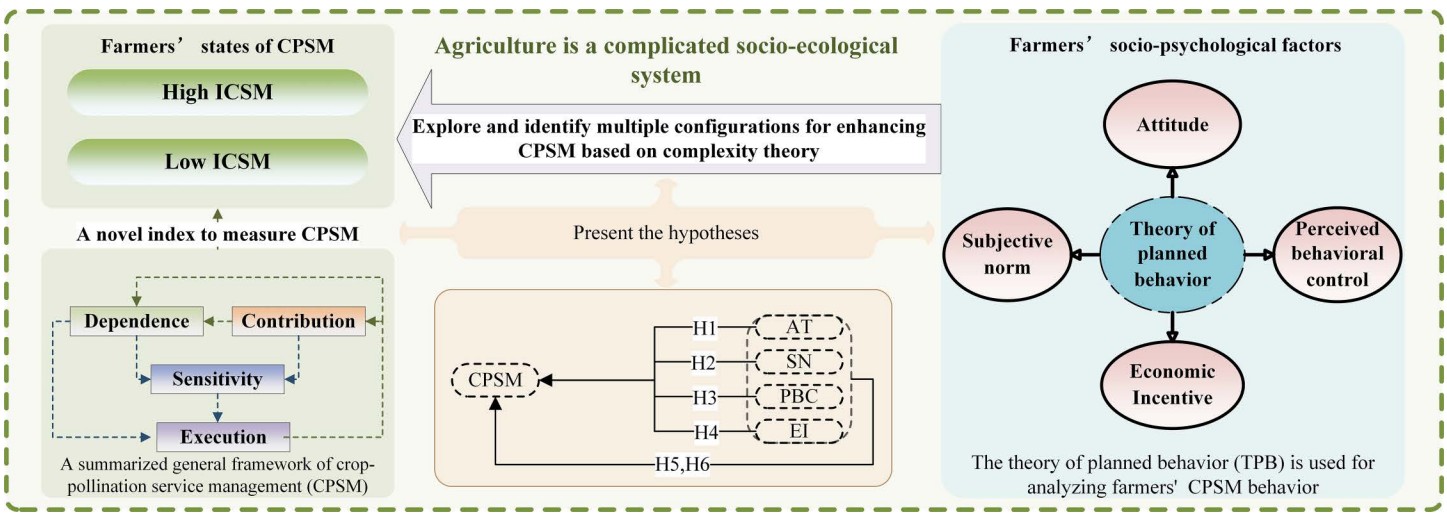

**Fig 2. Theoretical framework.**

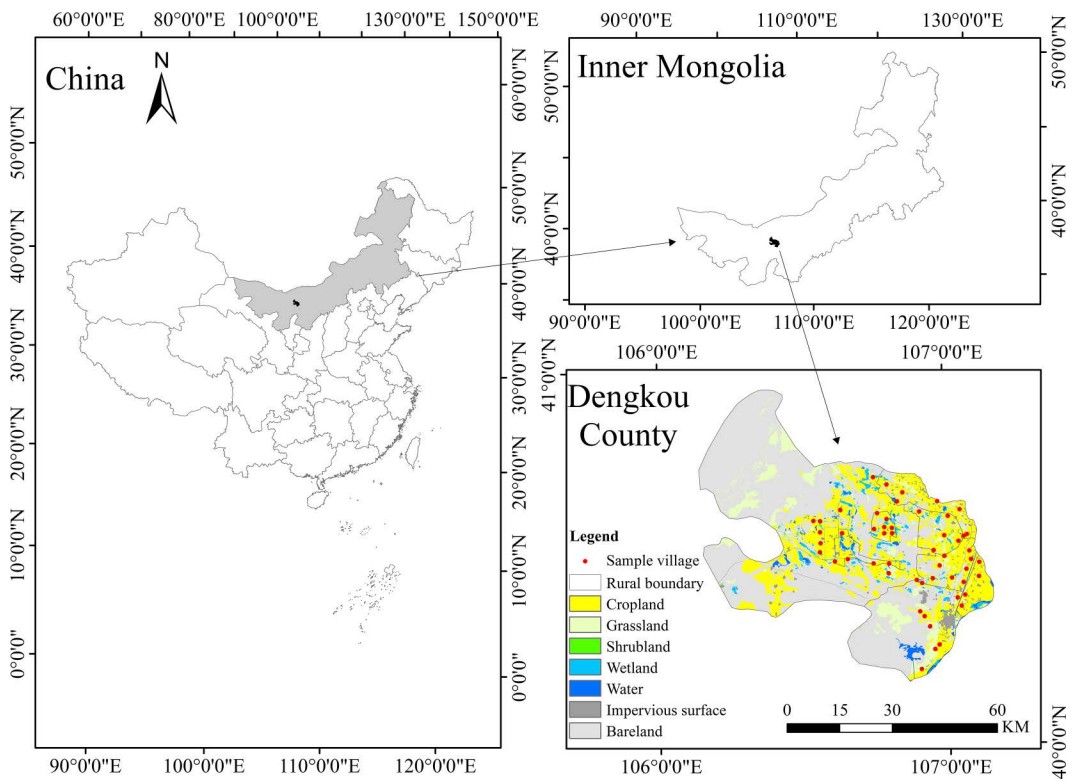

**Fig 3. Study region and sample village.**

106°09'-107°10'E and 40°09'-40°57'N. The region experiences a temperate continental monsoon climate with long cold winters, and hot and dry summers. It is a typical farming area and has a continuous history of agricultural cultivation throughout its long history. In Dengkou County, corn (account for 53.08%), sunflower (account for 31.8%), melons (account for 11,5%) and a few other crops (account for 3.62%) are grown. Among these crops, sunflower and melon rely more on pollination mediated by insects. Corn and sunflower bloom in July and August, while melons bloom in late spring and summer. Pollinators are mainly bees, butterflies and flies, of which there are two types, wild bees and domestic bees. However, policies and informal regulations in Dengkou County do not give sufficient attention to crop-pollination services, and farmers lack adequate awareness of CPSM. Therefore, the current conventional agricultural practices in the region (such as agricultural expansion and intensification) pose a serious threat to crop pollination services. Thus, researching on how to strengthen CPSM in study area offers interesting cases to developing CPSM and addressing above challenges.

### 3.2 Questionnaire and survey

We used a typical sampling approach to select 54 sample villages under sample towns and used a completely random sampling method to select 4–6 sample farmers from every target village. The locations of sample villages are shown by the red dots on the map of Dengkou County in Fig 2. The survey was conducted through random household visits in July and August 2021 by professionally trained researchers with ecological economics backgrounds. The recruitment period for this study started on **01/07/2021** and concluded on **31/08/2021**. Face-to-face interviews improve the accuracy of information obtained.

The Ethics Committee of the School of Economics and Management, Inner Mongolia Normal University, Hohhot, China, approved this study. All procedures involving human participants adhered to ethical standards, including the Declaration

**Table 2. Descriptive statistics of demographics.**

| Variable | Description | Min. | Mean | Max. | Std. Dev. |
|---|---|---|---|---|---|
| Gender | 0 = males, 1 = females | 0 | 0.33 | 1 | 0.47 |
| Education | 6 = primary school, 9 = junior high school, 12 = senior high school, 15 = junior college | 6 | 8.09 | 15 | 2.39 |
| Age | >0 year | 26 | 56.63 | 76 | 8.70 |
| Agricultural acreage | >0 hectare | 0.13 | 4.79 | 40.00 | 73.50 |
| EI | Agricultural profit ($) | 51.62 | 5293.50 | 76548.67 | 8236.26 |

of Helsinki. However, due to the nature and design of the study, this study did not involve human or animal experimentation which may raise ethical issues, and therefore no separate ethics approval code was applied. Informed consent was obtained from all participants, and data were collected securely via online software to protect personal information.

The research teams randomly interviewed peasant households in the study area to conduct questionnaire surveys. Participants were fully informed about the content and purpose of the questionnaire before deciding to participate. In addition, it is important to note that all participants were informed of their right to access information and were assured of the confidentiality of their personal data. We began the survey after obtaining informed consent forms completed by the participants.

A total of 267 completed questionnaires were returned, all of which were deemed valid. The questionnaire covered elements related to CPSM and TPB, with specific questions listed in S1 Appendix (Table 1: Weights of the principles and chosen elements used in the index; Table 2: Descriptive statistics of the elements used to measure TPB constructs). Other data are collected from the Dengkou County People's Government and the seventh population census. The data sources can be found in S3 Data.

### 3.3 The calculation method of the integrated index of CPSM

The restrictions of farmers' CPSM, to a large extent, is because of insufficient assessment methods. However, measuring farmers' CPSM, which should account for specific regions and various crop types, is challenging. Consequently, there is a need to develop an index based on the framework of CPSM for taking the local context into account. Here, we present an approach to measure farmers' CPSM based on the elements of four principles of the framework of CPSM. See S1 Appendix (Table 2) for meanings of elements. To calculate the scores for each principle, we summed the maximum points of the elements within each principle and computed the corresponding scores based on weighted sums of the selected elements (Eq. 1). This process involved calculating respondents' scores for dependence, contribution, sensitivity, and execution using weighted principles and elements (S1 Appendix: Algorithm for principles' scores). These scores were then aggregated to obtain the Primary index and subsequently converted to index's scores ranging from 0 to 100 (S1 Appendix: Algorithm for converting the *Primary_ICSM* into the integrated index).

$$Max\ score\ of\ principle_i = \sum_{j=1}^{n} W_{jimax} \tag{1}$$

*Max score of principle$_i$* is max score of the $i$th principle, $W_{jimax}$ is max weight of the $j$th element in the $i$th principle. See S1 Appendix (Table 1, column eight) for max weight of the $j$th element.

Since we mainly utilize the Likert scale for evaluation, different questionnaire elements are converted into scores of 1 (such as the 5-point system is 0, 0.25, 0.5, 0.75, 1). And dividing the score of each element by the max score of principle to avoid unduly emphasizing certain principles. By reason of assumption that some elements are more important than others in index, difference in practices' max/min weights is achieved by setting a weight for the min and max performance

of element. For example, respondents who reported performing an element at level 2 exercises would receive 0.75 times $W_{jmin}$ and 0.25 times $W_{jmax}$. Although linearity is a limitation, we find that the benefits of more complexity are too small to withstand the increased complexity.

To ensure the elements are relevant for Dengkou County, we reviewed literature, conferred with experts, and performed test interviews with members of the survey population. To determine the weights associated with principles and elements, a variant of the Delphi method known as Mini-Delphi (estimate-talk-estimate) was employed [64]. This approach facilitates a collective assessment of a predetermined set of questions by experts, allowing them to adjust their views through structured discussions. For this study, three experts with extensive knowledge of agricultural pollination in Dengkou County, specifically in pollination management, were consulted by the Dengkou County Agricultural Extension Service.

Detailed explanations of index's calculation methods can be found in S1 Appendix (The supplement of the integrated index's algorithm). To deal with important aspects of the complex relationships among the four principles in the framework of CPSM, the survey responses are used to calculate index scores by using a weighted summation of measured elements and principles. On the one hand, it can well address issues of weights and numbers of elements per principles of the framework of CPSM and can well summarize complex and multi-dimensional of farmers' CPSM on real landscapes. On the other hand, an assessment of farmers' CPSM purely relying on the hypothesized relationships among the four principles in the framework of CPSM (See Section 2.1) is highly unrealistic, given the hard-to-quantify cause-effect relationships among the principles and the hard-to-model nature of complexity aspects for certain principles.

We develop the integrated index of CPSM as a continuous variable only relative to itself. Another advantage of the index is that each principle contains different specific survey questions, rather than a single question being set to represent multi-principles. In addition, the weight of elements in each principle is accounted for to prevent the creation of an unfair advantage for one principle over others. However, depending on local situation and the goal of enhancing CPSM, some principles of index may be more important than others. The approach strengthens the social network of the integrated index. Index's algorithms based on the framework of CPSM can be generalized to other regions, but it is recommended to use similar methods to derive different elements and weights based on regional characteristics.

### 3.4 Data analysis

Validity refers to correctness of measurement, reliability is the consistency and stability of measurement results. Cronbach's α is used to assess the validity and reliability of questionnaire results. It can be argued that results have a good internal consistency if Cronbach's α is greater than 0.7 [65]. Exploratory factor analysis (EFA) is performed using Kaiser-Meyer-Olkin (KMO) and Bartlett's test of sphericity, respectively, to check power and applicability of factor analysis [66].

Once the set of reduced variables has been obtained through factor analysis, linear regression studies are conducted to analyze the relationships between different variables. The basic formula is shown in Equation 7:

$$ICSM_i = \alpha + \beta_1 AT_i + \beta_2 SN_i + \beta_3 PBC_i + \beta_4 EI_i + \sum_{n=1}^{m} \beta_{n+4} X_n + \mu \tag{7}$$

$i$ indicates different individuals. $X_n$ are controls, which include *age*, *gender,* and *education*. We performed regression analyses to test the effects of different factors on the integrated index of CPSM in presence of control variables. Control variables were not added to NCA-fsQCA, because their inclusion may complicate logical reasoning [67].

This study mines the complexity of CPSM antecedent from a holistic perspective and attempts to use fsQCA-NCA to explain how configurations of multi-factors contribute to expected CPSM. Unlike traditional quantitative research methods, qualitative comparative analysis (QCA) examines the causality of influencing factors and CPSM in terms of both necessary and sufficient causation [68]. Necessary causation means that the occurrence of CPSM relies on

antecedent causes, while sufficient causation suggests the effects of influencing factors on CPSM. Because sample data cannot be logically divided directly according to the criterion of "belonging or not" to the set, we employ fsQCA to deal with partial subordination and adequately match the research object based on extended TPB analysis [34]. While fsQCA can identify necessary causation, it only qualitatively states "whether the antecedent condition is necessary or sufficient for CPSM" and does not quantitatively reflect the degree of necessity. NCA approach is employed to analyze better quantitatively necessary and sufficient more effectively [35], therefore, the combination of NCA and fsQCA will be the best way to analyze configurations of farmers' social-psychology factors to strengthen CPSM [69]. FsQCA3.0 and R4.1.1 software is used to conduct NCA-fsQCA [70,71]. The original consistency threshold, PRI consistency threshold, and case frequency threshold are set to 0.8, 0.6, and 1, respectively [72]. Code used in this paper can be found in S2 Code.

Independent sample t-test is an available tool to clarify the differences between two groups [73]. To analyze configurations of CPSM more delicately, this study employs independent sample t-test to examine the differences between social-psychological factors and four principles of the framework of CPSM.

## 4. Results

### 4.1 Demographics and CPSM of respondents

Respondents' demographics are shown in Table 2. Since males make up the majority of labor force engaged in agricultural operations, farmers interviewed are mostly males. Respondents' educational levels are generally low, and most of respondents' education levels are primary and junior high school. Age structure of respondents is high, which indicates that population aging is gradually highlighted. The study sample is consistent with actual statistics of the seventh population census. The mean and standard deviation of agricultural acreage are 4.79 hectares and 73.50, respectively.

Fig 4 reflects that index's scores of respondents show a normal distribution. S1 Appendix (Table 1) shows the specific elements included in each principle and the average weights assigned to principles and practices by three pollination experts. For principle I (dependence), respondents believe that sunflower and watermelon are more dependent on pollinators than maize. For principle II (contribution), respondents observe a high frequency of various pollinators in their fields. For principle III (sensitivity), respondents believe that crop yield and quality are strongly influenced by CPSM. For Principle IV (execution), respondents' practices of installing beehives and avoiding pesticide spraying at flowering time are more highly implemented. However, practices related to establishing plant boundaries and protecting vegetation and wildflowers

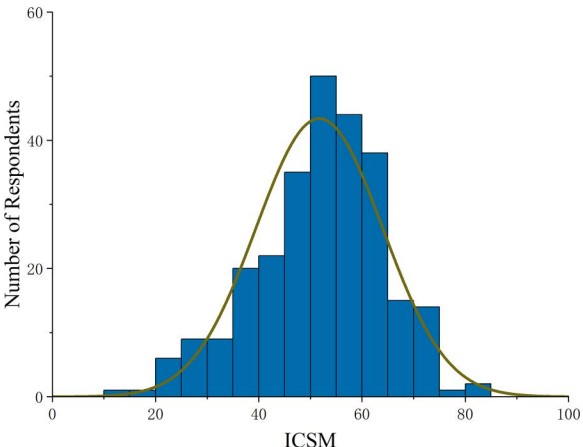

**Fig 4. Distribution of respondents' index scores.**

at farm boundaries are less implemented. Fig 5 reflects index's scores of respondents with different demographics. The results show that education and agricultural acreage are positively correlated with the integrated index of CPSM. Respondents aged 40–60 years have higher index compared to those under 40 years and over 60 years.

## 4.2 Measurement model test

The results of measurement model test are shown in S1 Appendix (Table 2). Cronbach'α is above 0.6, factor loading is above 0.6, and the overall Cronbach's α is 0.787. These indicate that indexes can be well represented by elements we selected, and extended TPB is reliable. The KMO is 0.773 and the *p*-value of Bartlett-test of sphericity is 0.000, which indicates the suitability of factor analysis.

## 4.3 Regression analysis

Fig 6 depicts the strength of relationships between the integrated index and explanatory variables. The correlation coefficients between the integrated index and *Education*, *AT*, *PBC* and ln*EI* are 0.16, 0.2, 0.12 and 0.21, respectively, indicating that the integrated index is positively correlated with Education, AT, PBC and EI. The correlation coefficient between the integrated index and SN is −0.17, indicating that the integrated index and SN are negatively correlated. Results of regression analysis are shown in Table 3. In model 1, we evaluate the baseline model, which included regression estimates for the integrated index and control variables only. Models 2–5 were created by adding independent variables to baseline model to examine how different indicators affect the integrated index, respectively. The inclusion of variables elevates explained variance of the integrated index. AT is positively correlated with the integrated index ($\beta=0.013$, $p<0.01$). Farmers with positive AT of CPSM tends to have higher CPSM levels, which is consistent with H1. SN is negatively correlated with the integrated index ($\beta=-0.013$, $p<0.01$). SN will have a negative effect on farmers' CPSM in Dengkou County, which is consistent with H2. PBC is positively correlated with the integrated index ($\beta=0.016$, $p<0.1$). Farmers with a stronger PBC of CPSM tended to exhibit higher CPSM levels, which is consistent with H3. EI is positively correlated with the integrated index ($\beta=0.055$, $p<0.01$). Higher EI positively supports higher farmers' CPSM levels, which is consistent with H4.

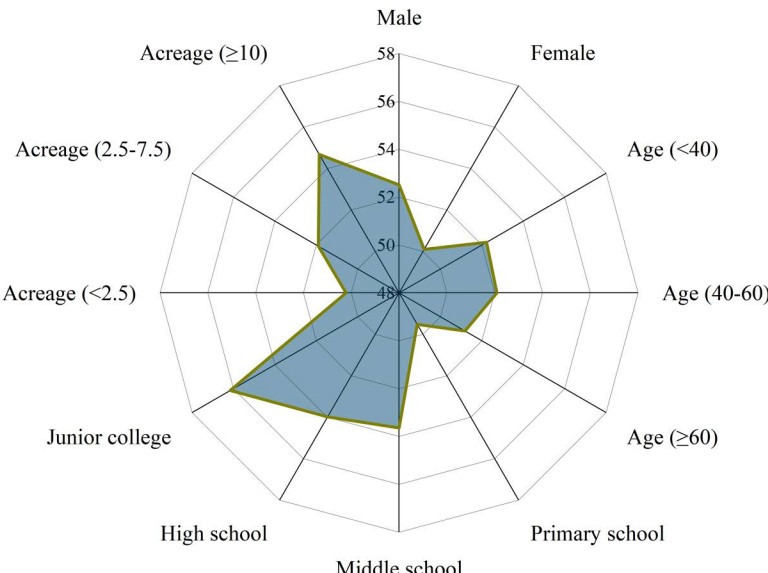

**Fig 5. Comparison of demographics and the integrated index of CPSM.**

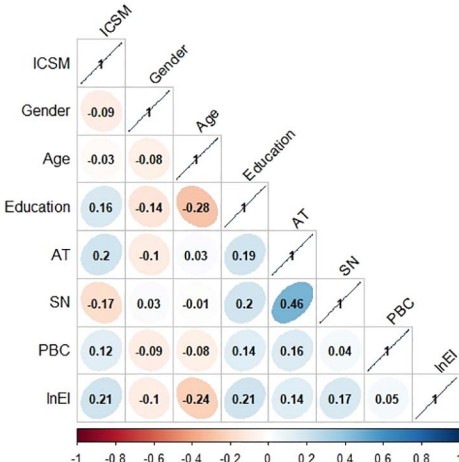

**Fig 6. Correlation matrix diagram of variables.** From −1 (red) to 1 (blue) indicates from perfect negative correlation to perfect positive correlation. The data within the graph are the values represented by the colors.

**Table 3. Results of regression analysis.**

| | Dependent variables: ln*ICSM* | | | | | |
| --- | --- | --- | --- | --- | --- | --- |
| | Model 1 | Model 2 | Model 3 | Model 4 | Model 5 | Model 6 |
| Controls | | | | | | |
| Gender | −0.048 (0.036) | −0.041 (0.036) | −0.042 (0.036) | −0.043 (0.036) | −0.035 (0.036) | −0.004 (0.034) |
| Age | 0 (0.002) | 0 (0.002) | 0 (0.002) | 0 (0.002) | 0.002 (0.002) | 0.002 (0.002) |
| Education | 0.018** (0.007) | 0.014* (0.007) | 0.023*** (0.007) | 0.017** (0.007) | 0.015** 0.007 | 0.015** (0.007) |
| Independent variables | | | | | | |
| AT | | 0.013*** (0.004) | | | | 0.022*** 0.005 |
| SN | | | −0.013*** (0.004) | | | −0.025*** (0.004) |
| PBC | | | | 0.016* (0.009) | | 0.012 (0.008) |
| lnEI | | | | | 0.055*** (0.015) | 0.060*** (0.014) |
| _cons | 3.772*** (0.148) | 3.692*** (0.148) | 3.807*** (0.146) | 3.553*** (0.187) | 3.269*** (0.195) | 3.011*** (0.210) |
| N | 267 | 267 | 267 | 267 | 267 | 267 |
| $R^2$ | 0.035 | 0.069 | 0.066 | 0.048 | 0.086 | 0.215 |
| Adj. $R^2$ | 0.024 | 0.055 | 0.052 | 0.034 | 0.072 | 0.194 |
| AIC | 68.996 | 61.590 | 62.313 | 67.375 | 56.659 | 21.827 |
| BIC | 83.345 | 79.526 | 80.250 | 85.312 | 74.596 | 50.525 |

Note: * $p < 0.1$; ** $p < 0.05$; *** $p < 0.01$.

In model 6 (Column 7 of Table 3), all independent variables are included. Variance inflation factors of all independent variables are less than 10, indicating that there is no complete multicollinearity. Result shows that the correlation between PBC ($\beta = 0.012$, $p > 0.1$) and the integrated index is not significant, which implies that there may be substitution or complementary effects between different factors in relational exchange, which supports H5. Therefore, we conduct NCA-fsQCA to explore potential configurations that influence CPSM.

## 4.4 NCA-fsQCA

**4.4.1 Data calibration.** Data need to be calibrated before using fsQCA and transformed into values between 0 and 1. Values in this study are fuzzy sets, so three limit values are used as the basis for calibration: 0.05 for complete disaffiliation; 0.5 for crossover; and 0.95 for complete affiliation [70]. Data calibration limit values are shown in Table 4.

**4.4.2 Necessity analysis.** This study examines the sufficiency and necessity relationships using fsQCA to test whether any single conditions are necessary for high or low integrated index. As shown in Fig 7, all conditions are below the consistency threshold of 0.9, indicating that no necessity conditions existed.

This study uses NCA as a complementary tool to fsQCA. NCA allows for the analysis of relationships between preconditions and necessity for different levels of outcomes (e.g., what is the minimum EI level for farmers to achieve the highest integrated index). The differences between NCA and fsQCA necessity analysis result in NCA identifying more necessity conditions than fsQCA. And two methods don't generate contradictions and can complement each other.

**Table 4. Calibration anchors of each fuzzy set.**

| Sets | Calibration | | | Descriptive analysis | | | |
|---|---|---|---|---|---|---|---|
| | Fully in | Crossover | Fully out | Mean | S. D. | Min | Max |
| ICSM | 70.33 | 53.25 | 28.16 | 0.51 | 0.29 | 0.01 | 1.00 |
| AT | 15.73 | 11.01 | 4.56 | 0.51 | 0.33 | 0.01 | 0.95 |
| SN | 14.39 | 6.92 | 3.02 | 0.46 | 0.35 | 0.03 | 0.96 |
| PBC | 13.93 | 13.92 | 9.29 | 0.79 | 0.32 | 0.00 | 0.95 |
| EI | 14007.67 | 2986.73 | 392.33 | 0.45 | 0.31 | 0.03 | 1.00 |

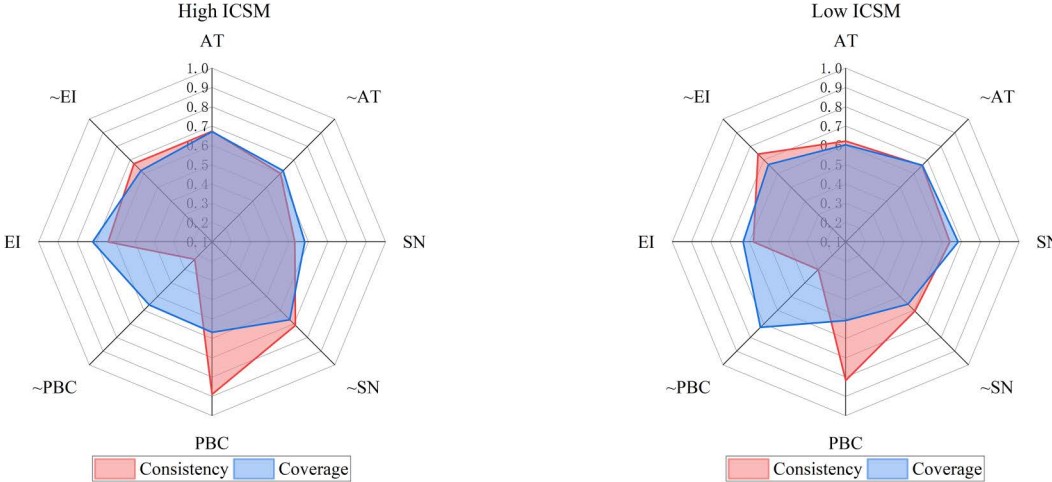

**Fig 7. FsQCA's necessity test for single conditions.** "~" denotes the absence of conditions.

Three lines are plotted above data points in XY scatter plot based on NCA: fitted curve (OLS), ceiling envelopment with a free disposal hull (CE-FDH) and ceiling regression with a free disposal hull (CR-FDH). The area in upper left corner of CE-FDH and CR-FDH relative to total area occupied by the sample reflects the degree of constraint of X on Y [74]. If the ratio of upper left corner is larger, the degree of constraint of X on Y is higher. As shown in Fig 8, the degree of constraint of precondition on the integrated index is weak. Whether a precondition is necessary mainly depends on: the effect size above the threshold of 0.1, and $p$-value$<0.05$. As shown in Table 5, all preconditions are less-than 0.1, so there is no necessary condition. In addition, Table 6 reports the results of bottleneck level analysis, where bottleneck level refers to the lowest level value (%) that a single precondition needs to satisfy within its range of observations to achieve certain level values (%) within the range of observations of results. If the integrated index reaches 100%, farmers' EI needs to reach 2.6% ($1,900.27).

Based on necessity analysis of fsQCA and NCA, there is no necessary condition for the integrated index.

**4.4.3 Configuration analysis.** In this study, fsQCA is utilized to analyze the grouping of conditions leading to high and low the integrated index. These different combinations of conditions denote configurations for achieving same results, respectively. This study uses nested results of simple and intermediate solutions to determine core conditions of each solution, with only the conditions of intermediate solutions being marginal conditions. The study focuses on configurations to enhancing CPSM. Due to the asymmetry of fsQCA causality, we can discuss the analysis of high and low integrated

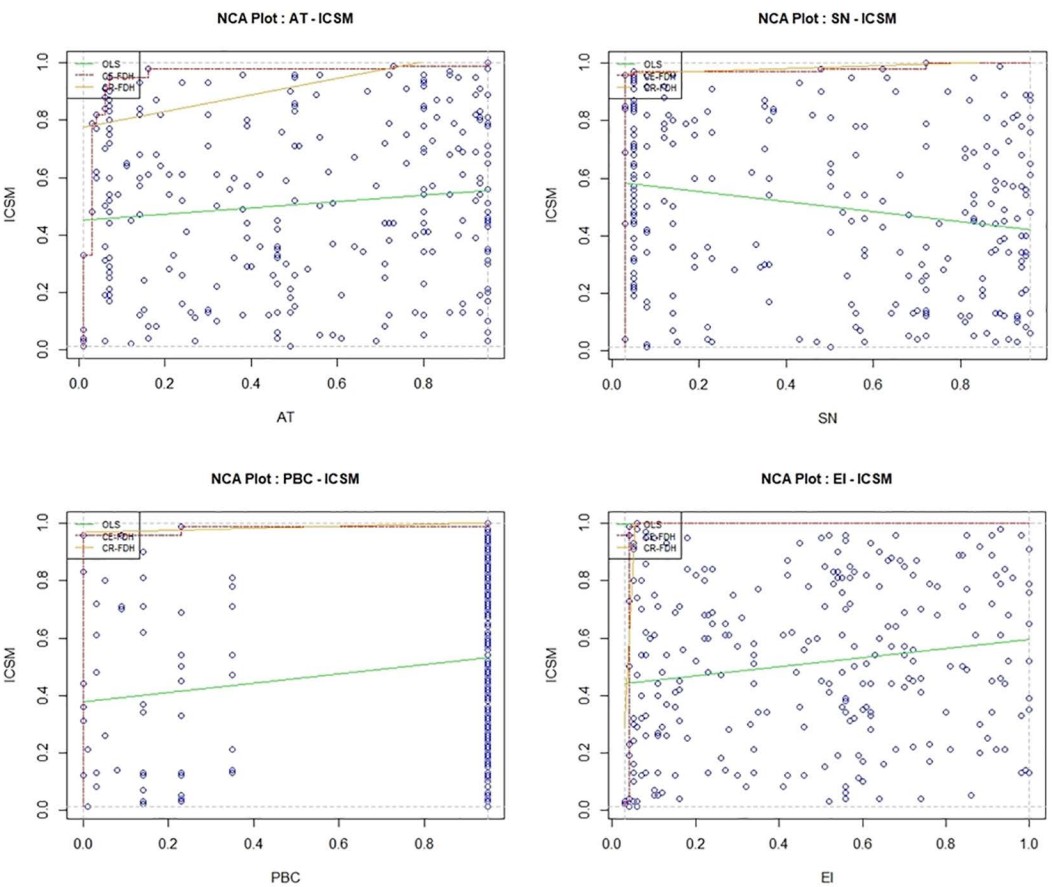

**Fig 8. Scatter plots with ceiling lines.**

**Table 5. Necessity analysis (NCA) for single conditions.**

| Condition | Method | C-accuracy | Ceiling zone | Scope | Effect size | *P*-value |
|-----------|--------|-----------|--------------|-------|-------------|-----------|
| AT | CR | 90.6% | 0.087 | 0.93 | 0.094 | 0.001 |
|    | CE | 100% | 0.038 | 0.93 | 0.041 | 0.004 |
| SN | CR | 99.3% | 0.014 | 0.92 | 0.015 | 0.360 |
|    | CE | 100% | 0.019 | 0.92 | 0.020 | 0.116 |
| PBC | CR | 99.6% | 0.013 | 0.94 | 0.014 | 0.773 |
|     | CE | 100% | 0.016 | 0.94 | 0.017 | 0.758 |
| EI | CR | 97.4% | 0.009 | 0.96 | 0.009 | 0.685 |
|    | CE | 100% | 0.010 | 0.96 | 0.010 | 0.577 |

**Table 6. Single conditional necessity bottleneck level.**

| PP | AT(%) | SN(%) | PBC(%) | EI(%) |
|----|-------|-------|--------|-------|
| 0% | NN | NN | NN | NN |
| 10% | NN | NN | NN | NN |
| 20% | NN | NN | NN | NN |
| 30% | NN | NN | NN | 0.0 |
| 40% | NN | NN | NN | 0.4 |
| 50% | NN | NN | NN | 0.8 |
| 60% | NN | NN | NN | 1.1 |
| 70% | NN | NN | NN | 1.5 |
| 80% | 10.0 | NN | NN | 1.9 |
| 90% | 46.2 | NN | NN | 2.2 |
| 100% | 82.4 | 81.4 | 91.0 | 2.6 |

Note: CR method, NN means unnecessary.

index to enhance results' generalizability. This study assumes that the conditions act on the integrated index regardless of their presence or absence. FsQCA analysis results are shown in Table 7. Three conditional groupings (CPSM1, CPSM2 and CPSM3) produced high integrated index with consistency indices of 0.832, 0.893 and 0.831, respectively, indicating that these conditional groupings are sufficient conditions for high integrated index. Moreover, the consistency of solution is 0.808, indicating that three conditional groupings covering most cases are also sufficient conditions for high integrated index. The coverage of model solution is 0.551, indicating that these three conditional groupings explain about half of high integrated index. There are also two conditional groupings with low integrated index: CPSM4 and CPSM5, with consistency of 0.879 and 0.902, respectively, an overall coverage of 0.870 for conditional groupings, indicating that both conditional groupings are sufficient conditions for low integrated index. These are consistent with H6.

**4.4.4 Robustness analysis.** This study selects to change the consistency threshold (adjusted from 0.8 to 0.85) to reprocess the sample data. The results indicate that configurations of high and low integrated index are identical to the subset of original results, and the resulting configuration is consistent, thus, the conclusion is stable.

## 4.5 CPSM's principles analysis based on configuration

To supplement the results of NCA-fsQCA further meticulously, independent sample T test is employed to examine whether there are differences in principles between high and low social-psychological factors, and results are shown in Table 8. Execution shows a difference between farmers with high and low AT. Farmers with high AT have higher implementation of CPSM. Execution and sensitivity show differences between farmers with high and low SN. Farmers with low SN in

**Table 7. Configurations analysis.**

| Antecedent variable | | High ICSM | | | Low ICSM | |
|---|---|---|---|---|---|---|
| | | CPSM1 | CPSM2 | CPSM3 | CPSM4 | CPSM5 |
| Ecological cognition | AT | ● | ● | | ⊗ | ⊗ |
| | SN | ⊗ | ⊗ | ⊗ | | ● |
| | PBC | ● | | ● | ⊗ | |
| Economic incentive | EI | | ● | ● | | ⊗ |
| Consistency | | 0.832 | 0.893 | 0.831 | 0.879 | 0.902 |
| Raw coverage | | 0.394 | 0.324 | 0.419 | 0.257 | 0.334 |
| Unique coverage | | 0.101 | 0.031 | 0.126 | 0.084 | 0.161 |
| Solution consistency | | 0.808 | | | 0.870 | |
| Solution coverage | | 0.551 | | | 0.418 | |

Note: ● = existence of core condition; ⊗ = loss of core condition.

**Table 8. CPSM's principles analysis based on configuration (Independent sample T test).**

| TPB | Principle | Mean ± standard deviation | | *t*-value | *p*-value |
|---|---|---|---|---|---|
| | | Low | High | | |
| AT | Execution | 5.12±0.69 | 5.27±0.64 | −1.88 | 0.061* |
| | Dependence | 4.54±0.65 | 4.60±0.65 | −0.74 | 0.457 |
| | Contribution | 5.31±0.66 | 5.41±0.57 | −1.35 | 0.178 |
| | Sensitivity | 4.21±0.65 | 4.22±0.55 | −0.13 | 0.894 |
| SN | Execution | 5.31±0.72 | 5.11±0.57 | 2.41 | 0.016** |
| | Dependence | 4.57±0.64 | 4.57±0.66 | 0.03 | 0.979 |
| | Contribution | 5.41±0.66 | 5.32±0.53 | 1.23 | 0.219 |
| | Sensitivity | 4.35±0.61 | 4.05±0.55 | 4.24 | 0.000*** |
| PBC | Execution | 4.98±0.73 | 5.27±0.64 | −2.85 | 0.005*** |
| | Dependence | 4.51±0.67 | 4.59±0.64 | −0.70 | 0.486 |
| | Contribution | 5.24±0.65 | 5.40±0.59 | −1.76 | 0.080* |
| | Sensitivity | 3.99±0.61 | 4.27±0.58 | −3.11 | 0.002*** |
| EI | Execution | 5.15±0.68 | 5.38±0.62 | −2.53 | 0.012** |
| | Dependence | 4.50±0.67 | 4.76±0.53 | −2.99 | 0.03** |
| | Contribution | 5.42±0.67 | 5.26±0.41 | 1.94 | 0.054* |
| | Sensitivity | 4.17±0.60 | 4.32±0.57 | −1.90 | 0.058* |

Dengkou county are more likely to appreciate the effects of CSPM on crop production, and are more willing to implement CPSM measures. Contribution, sensitivity, and execution show differences between farmers with high and low PBC. Farmers with high PBC have confidence in CSPM, place a greater emphasis on crop-production and have a higher level of practice. Dependence, contribution, sensitive, and execution show differences between farmers with high and low EI. Farmers with high EI have higher pollination trust and implement more CPSM.

## 5. Discussions

### 5.1 Farmers' demographics and CPSM

The descriptive statistics of the different principles of CPSM are shown in S1 Appendix (Table 3: Descriptive statistics of the different principles of CPSM), the small difference between the means of the different principles indicates that there

is little difference between the values of the different principles. Therefore, it is more valuable to explore relationships between demographics and the integrated index than to just compare scores of different principles. Fig 5 shows that the larger farming scales, the higher integrated index, i.e., agricultural acreage is positively correlated with the integrated index. Contribution and dependence of CPSM can be more clearly perceived by farmers during cultivation [75]. It may be that the larger area of crop cultivation, the more farmers attach importance to crop-pollination services, and therefore the easier it is to implement CPSM. However, excessive-scale farming tends to intensify agricultural production, and the resulting homogenization of agricultural landscapes will pose a serious threat to pollinator habitat management [76]. This broadly supports farmers' trust in the value of crop-pollination services, therefore, farmers with large planting areas can be guided to actively participate in CPSM to strengthen their efforts to implement pollinator-supporting practices, such as creating 'gardens' for bees at field boundaries [77]. Fig 6 shows that the correlation coefficient between education and the integrated index is 0.19, indicating that the effect of education on the integrated index is significantly positive, i.e., farmers with higher levels of education have higher integrated index. Well-educated farmers tend to have wider ranges of knowledge and are better able to understand CPSM [78]. Farmers with higher education often have stronger self-learning abilities and problem-solving abilities and are more likely to accept new technologies about CPSM, which is conducive to farmers to better coping with challenges arising from enhancing CPSM. Therefore, improving farmers' education level is an important strategy to enhance CPSM, and it is necessary to incorporate pollination knowledge into farmers' education and production.

**5.2 Configurations to enhance farmers' CPSM.**

To achieve enhance CPSM, farmers can via different configurations: AT-PBC path (CPSM1), AT-EI path (CPSM2), and PBC-EI path (CPSM3). It is noteworthy that the absence of SN is observed in all groups with high integrated index. The influence is reflected in that differences between dependence and contribution among farmers with high and low SN are not significant, and the sensitivity of farmers with high SN is lower than those with low SN. Farmers who are more affected by SN are less sensitive to the value of CPSM in regulating crop yield and quality. Therefore, SN does not support CPSM for farmers in Dengkou County. However, ignoring regional social context and simply reducing the role of SN on CPSM mean that a unified intervention strategy will not achieve the maximum efficiency of CPSM. CPSM is a kind of management with positive externalities, and external intervention should emphasize guiding farmers' external environment by formulating relevant policies to produce SN with positive externalities on CPSM. Based on three configurations to achieve better CSPM, the intervention strategy should design different optimal CPSM schemes for farmers' heterogeneity.

**AT-PBC path (CPSM1).** In CPSM, AT is indirectly reflected in farmers' evaluation of crop-pollination. Farmers with high AT values have stronger cognitions of the regulatory value of pollinators on crop-production, which directly affects the action orientation of farmers' CPSM [22]. In addition, high PBC not only provides confidence in executive for farmers' pollinator protection, but also can carry psychological construction of achieving better CPSM, so that the implementation of CPSM has room for continuous strengthening and improvement [79]. However, under conditions of high AT and high PBC, the absence of SN indicates that public service support has an irreplaceable role in promoting farmers to implement CPSM. For example, public information platforms such as "Agricultural Technology Promotion in China" not only provide real-time agricultural technology promotion but also publicize pollination knowledge [80]. The social effect caused by public service not only directly creates SN with positive externality, but also indirectly improves individuals' AT and PBC. Therefore, combination of high AT and high EI, with complete public service to support CPSM, constitutes a realization path of better CPSM.

**AT-EI path (CPSM2).** Farmers with high AT trust the value of crop-pollination services to agricultural production and ecosystems, so they are more likely to support the implementation of CPSM [25]. Agricultural benefits are a key attribute of EI, farmers with high EI are more sensitive to the positive impact of CPSM on crop-production [81]. Further, for farmers with both high AT and EI, pollinator protection is not only an environmental action, but also an important management to

increase agricultural production. Therefore, in terms of pollinator protection, high AT is the action orientation of high EI farmers, and high EI provides monetary gain for high AT farmers. However, in presence of high AT and EI, the absence of SN highlights the indispensable role of government support in promoting farmers to implement CPSM. For example, eco-compensation policies that provide positive externalities SN and economic support for pollinator protection can potentially improve farmers' AT about CPSM [82]. Therefore, combination of high AT and high EI, as well as effective pollination compensation policies, is an important way to enhance CPSM.

**PBC-EI path (CPSM3).** Farmers with high PBC tend to have stronger self-perceptions of practical ability and corresponding values of CPSM, which directly constitutes psychological construction about pollinator conservation trust [83]. Farmers with high EI have stronger capital input potential and agricultural output efficiency, and generally have more resources and funds to bear costs of implementing CPSM. Further, farmers with high PBC and EI are also more likely to seek education and training related to CPSM and invest in advanced technologies and equipment that support CPSM, and confidently apply knowledge and technology to implement CPSM based on strong sense of self-efficacy. These investments and practices can enhance both pollinator conservation and agricultural output benefits, ultimately becoming an important driver of a virtuous cycle of pollinator conservation and agricultural benefits. However, in presence of high PBC and EI, the absence of SN suggests that efficient market supports play an integral role in promoting farmers to enhance CPSM. In 2022, the difference between per capita income and expenditure of permanent rural residents in Dengkou County is $1,748 (source: Dengkou County 2023 government work report, available at http://www.nmgdk.gov.cn/zfxxgkdk/fdzdgknrdk/dkgzbg/202302/t20230206_499330.html), whereas the difference with the minimum EI required for the optimal level of CPSM is $152.27. Therefore, the disposable income of local farmers needs to be improved. Agricultural products premium about pollination can boost agricultural income and generate SN with positive externality for CPSM from market perspectives. Therefore, combination of high PBC and high EI, supported by effective market supports about pollination, is an important path to enhance CPSM.

### 5.3 Policy implications

#### 5.3.1 Motivating CPSM by enhancing farmers' attitude about pollinator.
In pursuit of pollinator protection in agriculture, government agencies can develop policies aimed at improving farmers' AT by promoting education and fostering supportive pollination cultures. Results suggest that farmers' willingness to adopt pollinator-friendly practices is influenced by education. So, governments worldwide should establish more agricultural education platforms (e.g., farmer field schools, science and technology academies) to improve farmers' awareness and AT towards crop-pollination and pollination practices [84]. The promotion of pollination practices should also be strengthened through official media and social platforms (e.g., new media of government affairs) to create a "social classroom" that complements education platform [85]. These recommendations are consistent with global practices, as agricultural education and public awareness campaigns have shown positive impacts in regions such as the European Union, where national governments have funded programs to promote pollinator conservation [86]. Similar initiatives in Latin America and Africa have demonstrated the effectiveness of farmer training programs in boosting adoption of sustainable practices [87].

#### 5.3.2 Innovating and popularizing CPSM technologies.
Farmers' mastery and application of advanced technology can enhance efficiency of implementing CPSM and increase their confidence and motivation to control pollination practices. The government should attach importance to the innovation capital investment of pollination technology, encourage technology-oriented enterprises to connect with farmers in order to solve practical technical problems faced by farmers in CPSM [88]. Focus on the research and development of CPSM's new agricultural mechanization equipment (e.g., agricultural drone assisted pollination), and increase innovative pollinator protection technologies (e.g., smart beehives) [89]. Establish information platforms related to pollinator protection to promote pollination technology, and guide farmers to use advanced CPSM technology through agricultural technology training [90]. Countries such as the U.S. and Canada have pioneered the integration of smart technologies into agriculture, leading to a significant increase

in pollination efficiency and farmers' engagement in pollinator protection strategies [91]. Similarly, China's recent push for technological innovation in agriculture has led to successful pilot programs involving drone-assisted pollination [92].

**5.3.3 Establishing "government + market" pollination compensation model to strengthen economic incentive.** Results show that, in execution principle, installing beehives, keeping their own bees or buying insect pollination services are not well implemented by farmers in Dengkou County. Government should establish pollination protection laws to provide legal protection for crop-pollination services, and the key is to clarify compensation objects (e.g., farmers, beekeepers, merchants), compensation contents (e.g., beekeeping and purchase of pollination services), and how to compensate (e.g., direct subsidies, pollination transition and maintenance subsidies) [93]. The enforcement agencies responsible for pollination protection should be strengthened to ensure strict adherence to pollination protection laws, and coordination between national legislation and local departments should be improved to support effective implementation [94]. Different pollination compensation standards should be set in different regions. Pollination compensation standard in Dengkou County should be $152.27, which is the lowest EI ($1900.27) of the optimal CPSM minus the difference between farmers' income and expenditure. Market-oriented, pollination compensation in market gradually transitions from price support to direct payment of consumer [95]. Insect-pollination certification can be provided to agricultural products that meet standards to increase consumer trust [96]. Farmers need to ensure the quality of insect-pollinated agricultural products and enhance consumers' value recognition through marketing strategies (e.g., packaging and branding of insect-pollinated agricultural products) [97]. Compared to existing policies for the protection of pollination services in the EU and the US, this policy implication allows for a more flexible and regionally targeted approach, tailored to local economic conditions and pollination needs, while ensuring a gradual shift from government-supported subsidies to market-driven incentives that have the potential to contribute to long-term sustainability and greater farmer participation [98].

## 6. Conclusion

Current agricultural intensification and increased agricultural expansion have caused a serious loss of crop-pollination services, which causes great threats to food security and agricultural sustainability. Enhancing farmers' CPSM is considered a significant option for keeping human well-being and a necessary way to promote sustainable development of agriculture. Previous studies on pollination management have been based on questionnaire descriptions of single practice, no scholars have constructed a comprehensive measurement of pollination management. This study innovatively proposes a framework of CPSM and constructs a continuous-type numerical integrated index to quantify farmer's CPSM in regional context, and analyzes relationships between the integrated index and demographics. Results show that farmers have high integrated index, and the integrated index is positively correlated with education and agricultural acreage. To further investigate configurations that strengthen CPSM, this study applies TPB extended from EI perspective to pollination studies for the first time. Based on complexity theory and results of regression analysis, possible complementarities or interchangeabilities among influencing factors are identified. Therefore, this study innovatively uses NCA-fsQCA for the first time in the field of agricultural management to explore multi-configurations to enhance CPSM. Compared with econometric models, NCA-fsQCA can be used to identify multi-configurations that strengthen CPSM, rather than the single solution reported by simple regression analysis. It is found that there are three configurations of high integrated index: AT-PBC path, AT-EI path, and PBC-EI path, and two configurations of low integrated index. The bottleneck table (Table 7) may help to make optimal policy allocation decisions, where the optimal state of CPSM requires at least EI $1900.27. Independent sample T test is used to analyze differences in principles between high and low social-psychological factors, providing in-depth analyses and supplements to the results of NCA-fsQCA. Our findings expand ecological literatures on farmers' social-psychology and CPSM, provide new strategies for exploring ecological economics from complexity theory in economics, and highlight the importance of multi-factors interactions in the formation of CPSM. Ecological literatures can inform and shape government policy by increasing trust in CPSM and offering an outline for understanding complex relationships between farmers' social-psychology and implementation CPSM.

## Supporting information

**S1 Appendix. Statistical results and the algorithm of the integrated index.**
(DOCX)

**S2 Code. R code used in this article.**
(DOCX)

**S3 Data. Data used in this article.**
(XLSX)

## Acknowledgments

The authors are grateful to the experts who participated in the survey and the residents for their cooperation and patience in the questionnaire survey. The authors thank the anonymous reviewers for their help in improving this paper.

## Author contributions

**Conceptualization:** Hongkun Zhao.

**Data curation:** Hongkun Zhao.

**Formal analysis:** Hongkun Zhao, Yaofeng Yang, Yajuan Chen.

**Funding acquisition:** Hongkun Zhao, Yajuan Chen.

**Investigation:** Hongkun Zhao, Yajuan Chen.

**Methodology:** Hongkun Zhao.

**Software:** Hongkun Zhao.

**Supervision:** Yajuan Chen.

**Writing – original draft:** Hongkun Zhao.

**Writing – review & editing:** Hongkun Zhao, Yaofeng Yang, Yajuan Chen, Huyang Yu, Zhuo Chen, Zhenwei Yang.

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
