## [Decision Letter · Decision Letter 0]

Dear Dr. Chen,

Thank you for submitting your manuscript to PLOS ONE. After careful consideration, we feel that it has merit but does not fully meet PLOS ONE’s publication criteria as it currently stands. Therefore, we invite you to submit a revised version of the manuscript that addresses the points raised during the review process.

We look forward to receiving your revised manuscript.

Kind regards,

Munir Ahmad, PhD

Academic Editor

PLOS ONE

Journal requirements: When submitting your revision, we need you to address these additional requirements. 1. Please ensure that your manuscript meets PLOS ONE's style requirements, including those for file naming. The PLOS ONE style templates can be found at https://journals.plos.org/plosone/s/file?id=wjVg/PLOSOne_formatting_sample_main_body.pdf and https://journals.plos.org/plosone/s/file?id=ba62/PLOSOne_formatting_sample_title_authors_affiliations.pdf. 2. Please note that PLOS ONE has specific guidelines on code sharing for submissions in which author-generated code underpins the findings in the manuscript. In these cases, we expect all author-generated code to be made available without restrictions upon publication of the work. Please review our guidelines at https://journals.plos.org/plosone/s/materials-and-software-sharing#loc-sharing-code and ensure that your code is shared in a way that follows best practice and facilitates reproducibility and reuse. 3. We note that the grant information you provided in the ‘Funding Information’ and ‘Financial Disclosure’ sections do not match.  When you resubmit, please ensure that you provide the correct grant numbers for the awards you received for your study in the ‘Funding Information’ section. 4. Thank you for stating the following financial disclosure:  [Yajuan Chen: National Natural Science Foundation of China (No. 32060317) , the Inner Mongolia Natural Science Foundation (No. 2023MS04014), and The Fundamental Research Funds for the Inner Mongolia Normal University (No. 32150022210).].  Please state what role the funders took in the study.  If the funders had no role, please state: ""The funders had no role in study design, data collection and analysis, decision to publish, or preparation of the manuscript."" If this statement is not correct you must amend it as needed. Please include this amended Role of Funder statement in your cover letter; we will change the online submission form on your behalf. 5. Thank you for stating the following in the Acknowledgments Section of your manuscript: [The authors gratefully acknowledge the financial support from the National Natural Science Foundation of China (No. 32060317) , the Inner Mongolia Natural Science Foundation (No. 2023MS04014), and The Fundamental Research Funds for the Inner Mongolia Normal University (No. 32150022210). The authors are grateful to the experts who participated in the survey and the residents for their cooperation and patience in the questionnaire survey. The authors thank the anonymous reviewers for their help in improving this paper.]We note that you have provided funding information that is not currently declared in your Funding Statement. However, funding information should not appear in the Acknowledgments section or other areas of your manuscript. We will only publish funding information present in the Funding Statement section of the online submission form. Please remove any funding-related text from the manuscript and let us know how you would like to update your Funding Statement. Currently, your Funding Statement reads as follows:  [Yajuan Chen: National Natural Science Foundation of China (No. 32060317) , the Inner Mongolia Natural Science Foundation (No. 2023MS04014), and The Fundamental Research Funds for the Inner Mongolia Normal University (No. 32150022210).].   Please include your amended statements within your cover letter; we will change the online submission form on your behalf. 6. Please include captions for your Supporting Information files at the end of your manuscript, and update any in-text citations to match accordingly. Please see our Supporting Information guidelines for more information: http://journals.plos.org/plosone/s/supporting-information. 

Additional Editor Comments:

Need to focus the point-wise improvement of the manuscript especially from the reviewer with rejection

Reviewers' comments:

Reviewer's Responses to Questions

**Comments to the Author**

1. Is the manuscript technically sound, and do the data support the conclusions?

Reviewer #1: Partly

Reviewer #2: Yes

Reviewer #3: Yes

2. Has the statistical analysis been performed appropriately and rigorously?

Reviewer #1: No

Reviewer #2: Yes

Reviewer #3: Yes

3. Have the authors made all data underlying the findings in their manuscript fully available?

Reviewer #1: No

Reviewer #2: Yes

Reviewer #3: Yes

4. Is the manuscript presented in an intelligible fashion and written in standard English?

Reviewer #1: No

Reviewer #2: Yes

Reviewer #3: No

Reviewer #1: The authors focus on the important role of pollination in ecosystems and also note that pollination systems are important for sustainable development and securing crop production and food supply. The research extends ecological research on farmers' psychosocial and crop pollination service management, provides new strategies for exploring ecological economics from the complexity theory of economics, and highlights the importance of multifactorial interactions in shaping crop pollination service management (CPSM). The authors seek to inform government policy development, increase confidence in crop pollination service management, and provide a framework for understanding the complex relationship between farmers' social psychology and the implementation of crop pollination service management.

In the study, the authors presented the current situation of crop pollination-related systems in Dengkou County, analysed the role of government management, farmers and other relevant factors in crop pollination-related work, and concluded that strategies to promote the management of crop pollination services should take into account farmers' perceptions, knowledge and roles in improving pollination, and that the level of education and agricultural area are positively related to the management of crop pollination services. However, the study only analysed the impact of farmers on the functioning of crop pollination systems without understanding the underlying logic of pollination systems, and did not take into account the natural ecological environment, anthropological farming practices, changes in crop species, background and changes in pollinator diversity, and ecological factors affecting pollination as directly relevant to the pollination system, and lacked analysis of the basic knowledge of the underlying biological systems to support it. There is also no analysis of the impact on pollinators of a major human action - changes in large-scale, man-made agriculture.

1. introduction should be supplemented with the type and species of pollinators, the part did not pay attention to the flowering period of crops, other plants and pollinators change information, this part of the content is closely related to pollination, it is the main factors that lead to anthropogenic influence on pollination system, these contents will lead to changes in the concept of pollination system of farmers of the main content. This part was missing in the study, which may directly lead to the wrong direction of the study.

2. Research should also focus on the diversity and variability of flowering times and pollinators of etiolated crops and other plants, and the reasons for this variability.

3. Study should complement the development history, current status and importance, differences, advantages and disadvantages, development trends and problems of traditional natural crop pollination, crop pollination in industrial agriculture and crop pollination by insects and other professional pollinators. Based on the results of this study, what measures should be taken to protect the way pollinators pollinate crops?

4.There are flaws in the content of the study that lead to problems with the design of the study. ‘Contribution is the frequency of occurrence of farmers observing pollinator visits to the crop’, this part of the study is inadequate in terms of the contribution of the pollinators and inadequate for the education of the farmers. This is a flaw in the overall study, and the success and significance of the study is directly determined by the correct and sound choice of research components.

5. Methodological component: In Dengkou County, maize (53.08%), sunflower (31.8%), melon (1.5%) and a few other crops (3.62%) are grown. Among these crops, sunflower and melon are more dependent on insect pollination. The lack of information on changes in the flowering period and pollinators of these crops and other plants closely related to pollination are the main factors leading to anthropogenic influences on the pollination system, and even more to changes in farmers' perceptions of the pollination system.

Reviewer #2: The manuscript entitled “Configuration analysis of crop-pollination service management: a novel insight from the theory of planned behavior” proposed a framework and introduced the quantitative integrated index to evaluate the crop-pollination service management. Relating to socio-psychological factors, the authors attempt to explain the underlying mechanisms of farmers’ crop-pollination service management in the regional context. This manuscript addressed the importance of improving farmers’ participation in the pollination management, and provided useful information for areas of crop production, pollination ecology, and ecological economics.

However, there are several aspects in the manuscripts needs to be improved before it is suitable for the publication. With regard to the manuscript structure, the first two sections accounted for one third of the manuscript, which should be shortened. Many sentences were repeated throughout the text. There are many indications in the results part, but a lack of supporting references in the discussion, especially in 5.4. Discussions at a broader scale are also needed. Please see the detailed comments below:

1. Introduction

Introduction can be more concise, by focusing on the introduction of CPSM, the current limitations in measuring CPSM, following by introducing the concept of TPB and the research gaps. The text should be shortened, especially when section 2 provided more detailed background and hypotheses, and both sections reached to more than 30% of the manuscript. Some of the text can be moved to the Appendix.

There are many abbreviations in the text, the authors are recommended to reconsider the usage of abbreviations instead of introducing too many new abbreviations, e.g., based on the first abbreviation CPSM, the ICSM can be written as “integrated index of CPSM”, and FCPM can be written as “framework of CPSM”, etc.

L50: “about 75% of global crops” varied in their dependence on pollinators, please check the review: “Importance of pollinators in changing landscapes for world crops” from Klein et al. (2007) and add the citation.

L80: farmer behavior � farmers’ behavior

L86-94: these sentences can be shortened.

L82, 86: FCPM or FCPSM? As in L64, CPSM was the abbreviation of crop-pollination service management. It can be written as “framework of CPSM”.

L101: “Index of crop-pollination service management” could be written as “index of CPSM” instead of ICSM and why not use “ICPSM”? Please reconsider the abbreviations used in the text. It can be confusing for readers with many abbreviations in different forms.

L131-132: “fuzzy set quantitative comparative analysis” can be deleted, as the “fsQCA” was explained in L129.

L138-139: “crop-pollination service management” � “CPSM”.

L139-140: please add references for this example.

L147: delete the extra “Knapp”.

L148-149: the abbreviation ICSM refers to “integrated index of crop-pollination service management” in this objectives, is the “integrated” included in the abbreviations? Or it can be written as “integrated index of CPSM”.

2. Theoretical background and hypotheses

L157: add “of” after “framework”.

L168: at least tomato and sunflower are not only relied on insect pollination, they also have the ability of self-pollination, it is inappropriate to use “complete dependence” for these plants. Maybe use partially dependence, and please add references for the plant examples.

L169: “plant breeding systems” may be a more suitable phrase instead of “plant physiology”.

L169: add “about” or “on” after “farmers’ observation”.

L169-170: Please specify “the frequency of occurrence about pollinator’s visits”, does it mean that the relative abundance of different pollinator species, or specific pollination behaviour such as visitation rate, foraging speed, etc.

L171: add “on” after “context-dependent”?

Fig.1: The figure itself should be self-explanatory, e.g., the specific meanings of A-F arrows should be indicated clearly. Appropriate terminologies or specific terms can be used here instead of using A-F. The limited contributions from the pollinators due to lack of pollinators or effective pollination behaviour may influence the sensitivity. However, if principle II refer to the pollinators’ visits to crops (L165-166), then the meaning of arrow C can be confusing when the supplementary pollination technique was added to the process. Please specify where this additional pollination technique located in the figure. Are there any other arrows that were not presented in the figure? For example, contribution may directly influence execution, farmers may adopt pollinator-related management before there were effects of yield and quality. When there were only a few bees available in the field, farmers may bring more bee hives before finishing of flowering.

L179: add “of” after “adoption”.

L197: add “of the” after “complexity”.

L200-201: are there any references available for this statement?

L204: government � governmental

L205: please give a short description of Dengkou County, when it appears for the first time in the text.

L208: what is “EI” stands for?

L278: add “a” after “used”.

L285-292: is this part belong to the Ethics Statement?

L296-297: this part was repeated in L283-284.

L297-300: this part was repeated in L291-294.

L308: add “of” after “measurement”.

L321, 325, 329, 333: “scorce” � “scores”.

L327: column 7 and 8?

L390: add references for the two software used.

L393: why use Mann-Whitney U test here? It is used for nonparametric datasets. There are many tests to compare the differences for two groups. The reason for using this test is insufficient here. Please reconsider the selection of tests and specify the reasons for using the test.

4. Results

The indications and citations should not be included in the Results part (e.g., L463, 471, section 4.4.4, etc.), they should be moved to the Discussions. Also, the reasons for using a specific analytical tool should be in the Methodology, please check if there is a repeated information between Methodology and Results.

L404: when present mean values in the text, standard deviations or standard errors should be added.

L405: the EI values were already presented in Table 1.

L413: sensitive � sensitivity

L414: executive � execution

L442: add a space between “ofTable2”.

Figure 6, Table 7: the abbreviations shown in the figure and table should be explained in the legend. There is a lack of description and explanations of colored values of -1 to 1. Also, there is a lack of explanations of the values in the table and parenthesis, respectively.

5. Discussions

In general, there were many repeated information and descriptions in the discussions compared to the previous text, the authors are recommended to check throughout the text. The indications presented in the discussion should attach the relevant statistical results, and refer to a specific figure and table. Many of the statements and speculations in the discussion are lack of supporting references. The implications should also be extended to a broader scale, i.e., more references and/or comparisons of other studies at the national level or global levels are needed.

L535-540: this part was repeated in the introduction L96-99. The section 5.1 is more like the reasons for choosing the analytical tools and can be moved to the methodology.

L 552: please specify what “little difference” means by providing specific statistical results.

L554-555: when present results in the discussion, the relevant statistical results should be presented as well, referring to the specific table or figure.

L564: specific statistical results should be presented here.

L575-578: this part was mentioned previously.

L583: which statement is this reference referred to?

L590: the definition of AT was repeated.

L599-601, 629-631, 666-668: please add references to these statements and examples.

Section 5.4: there is a lack of references in this sections, please add references for many of the examples in this section.

Reviewer #3: Comments-RGB

Positives

The study demonstrates a high degree of analytical rigor through its use of multiple advanced statistical and analytical techniques, showcasing the robustness of its methodology. The combination of regression analysis, necessary condition analysis (NCA), fuzzy-set qualitative comparative analysis (fsQCA), and the Mann-Whitney U test provides a multi-faceted approach to examining the factors influencing crop-pollination service management (CPSM). Each method contributes unique insights, enhancing the reliability and depth of the findings.

Regression analysis effectively identifies direct relationships, revealing that education level and agricultural acreage are positively correlated with CPSM. This establishes a clear statistical foundation for understanding key influencing factors. The inclusion of NCA ensures that essential conditions for CPSM are identified, providing a more nuanced perspective on what factors are indispensable. FsQCA adds a configurational perspective, allowing the study to explore combinations of conditions that drive CPSM behaviors, which is particularly valuable for understanding complex, multi-causal phenomena. The Mann-Whitney U test further strengthens the analysis by comparing differences between groups, ensuring the findings are robust across various farmer demographics.

By integrating these diverse analytical approaches, the study not only corroborates its findings across multiple methods but also provides a comprehensive understanding of the dynamics at play. This robust analytical framework enhances the validity of the study’s conclusions and supports its practical implications for improving CPSM among farmers. The depth and breadth of the analysis reflect a well-executed research design, making it a significant contribution to the field of sustainable agriculture.

Need for further considerations

However, the study requires major review and some corrections before consideration can be given for publication. These re outlined below:

Suggested issue 1:

The study provides a comprehensive exploration of crop-pollination service management (CPSM) and identifies critical factors influencing farmers’ behaviors. It employs a robust methodological framework, grounded in the Theory of Planned Behavior (TPB), and uses a range of analytical techniques such as regression analysis, necessary condition analysis, fuzzy-set qualitative comparative analysis (NCA-fsQCA), and the Mann-Whitney U test. The research reveals that education level and agricultural acreage positively correlate with CPSM and identifies three causal configurations for enhancing CPSM: the AT & PBC path, the AT & Economic Incentive path, and the PBC & Economic Incentive path. Additionally, the study finds contrasting effects of antecedent variables across different principles of CPSM and establishes that the optimal state of CPSM requires an economic incentive of at least $1900.27. These findings offer practical strategies for improving CPSM and increasing farmer participation in pollinator-supporting behaviors in agricultural practices.

However, the study does not address cultural backgrounds (including local traditions and religion), which are often pivotal in shaping farmers' perceptions, knowledge, and decisions about agricultural practices. Cultural values, traditions, and norms can significantly influence attitudes (AT), perceived behavioral control (PBC), and economic considerations, which are central to the TPB framework used in this research. The omission of cultural factors may limit the depth and applicability of the findings, especially given the rural context of Dengkou County, where traditional beliefs and community practices likely play a role in shaping agricultural behavior.

Explore cultural and regional factors that might influence the applicability of the findings to other settings beyond Dengkou County. To enhance the study, it would be valuable to integrate an analysis of cultural influences on CPSM behaviors. For instance, exploring traditional agricultural practices, beliefs about pollination, and local community norms could provide a richer understanding of farmers’ decision-making processes. Incorporating cultural variables into the survey or conducting qualitative interviews to examine these aspects would add depth and nuance to the findings. Such an approach would contextualize the results within the socio-cultural environment of Dengkou County, making the study more robust and actionable. By addressing the role of cultural background, the research could offer a more comprehensive perspective on the factors driving CPSM, ultimately enhancing its theoretical and practical contributions to sustainable agriculture.

Suggestion issue 2:

While the study focuses on socio-economic and behavioral factors, it could benefit from integrating ecological data, such as the diversity and abundance of pollinators in the surveyed farms, to correlate management practices with actual pollination outcome. The study is well-structured and provides valuable insights into CPSM. By incorporating ecological factors, validating tools, and exploring regional differences, the analysis can become even more robust and impactful. These suggestions can enhance the study’s ability to inform effective CPSM strategies and policies.

Suggestion issue 3:

A cross-sectional survey provides a snapshot, but a longitudinal approach could better capture changes in farmer behavior over time and the long-term impact of interventions. This may relooked at or justification for exclusion provided

Suggestion issue 4:

The mention of consulting regional pollinator experts and non-academic stakeholders in the discussion section, rather than the methodology, is a notable issue. It creates a gap in the transparency of the research process. If these consultations played a role in constructing the framework, they should have been explicitly described in the methodology to clarify how these insights influenced the study's design.

To improve, the consultation process should be added to the methodology section, detailing how experts and stakeholders contributed to framework development. Additionally, the influence of these consultations on the research outcomes should be explicitly stated, ensuring the study reflects a comprehensive and well-informed approach to the ecological context.

I strongly recommend relocating the following section from the supplementary materials to the methods section, as it is essential to the study and directly supports the extensive deliberations presented in the discussion and also presented in data analysis:

"To ensure the elements are relevant for Dengkou County, we reviewed literature, conferred with experts, and performed test interviews with members of the survey population. To determine the weights associated with principles and elements, a variant of the Delphi method known as Mini-Delphi (estimate-talk-estimate) was employed (Pan et al., 1996). This approach facilitates a collective assessment of a predetermined set of questions by experts, allowing them to adjust their views through structured discussions. For this study, three experts with extensive knowledge of agricultural pollination in Dengkou County, specifically in pollination management, were consulted by the Dengkou County Agricultural Extension Service."

Including this section in the methods would provide much-needed clarity on how the principles and weights were derived, which is central to the study's framework. Furthermore, its prominent role in the discussion underscores its importance, making its inclusion in the methods section critical for readers to fully understand and contextualize the findings. This move would enhance the study’s transparency, methodological rigor, and alignment between the methods and discussion sections.

Suggested issue 5: Critique of placement in main text vs. supplementary material

1. Relevance to core methodology

Equations like these are critical to understanding the methodology used to calculate scores and indices central to the study. They detail how the analysis is conducted and provide transparency, ensuring replicability. However, they are dense and mathematical, which might distract readers from the broader methodological narrative if presented in the main text.

2. Purpose of the equations

o If the equations are essential for readers to follow and evaluate the results and methodology, they should remain in the main text.

o If they are primarily supporting details for replicability or validation purposes, they could be moved to supplementary material.

3. Accessibility for the audience

o For a broad audience, including practitioners and policymakers, presenting such detailed equations in the main text might reduce readability.

o For a more specialized audience, such as academic researchers familiar with these methods, keeping them in the main text might be justified.

4. Current context

o The equations are highly detailed and seem to be supported by a reference to the appendix, where additional data and variables (e.g., weights, algorithms) are described. This redundancy suggests they could be summarized in the main text with a clear reference to supplementary material for those seeking further detail.

Recommendation: balanced approach

• Main Text: Provide a concise summary of the methodology, outlining the key steps in plain language. Include one representative equation (e.g., Eq. 1) to illustrate the process, but avoid including all equations.

• Supplementary Material: Move the detailed equations (Eqs. 2-6) and algorithmic descriptions to the supplementary material, as they are technical details that support the core methodology but are not critical for the broader understanding of the study.

Suggested rewrite for main text

In the main text, you could say:

"To calculate the scores for each principle, we summed the maximum points of the elements within each principle and computed the corresponding scores based on weighted sums of the selected elements (Eq. 1). This process involved calculating respondents' scores for dependence, contribution, sensitivity, and execution using weighted principles and elements (detailed in Supplementary Material, Eqs. 2-6). These scores were then aggregated to obtain the Primary ICSM and subsequently converted to ICSM scores ranging from 0 to 100 (see Supplementary Material for the complete algorithm)."

Suggested issue 6:

The authors introduced the Theory of Planned Behavior (TPB) and its extension using economic incentives (EI) as a framework for analyzing crop-pollination service management (CPSM). However, their explanation lacks a thorough introduction and context for readers unfamiliar with TPB. The explanation of the Theory of Planned Behavior (TPB) in the manuscript is inadequate, lacking a thorough introduction to its core concepts. Key components like attitude (AT), subjective norm (SN), and perceived behavioral control (PBC) are briefly mentioned without sufficient context or real-world examples, making it inaccessible to readers unfamiliar with the theory. Additionally, the manuscript introduces complexity theory in the section title but fails to elaborate on it, causing confusion. The inclusion of economic incentives (EI) to extend TPB is not adequately justified, as assumptions about farmers’ motivations lack supporting evidence. The section's structure is disorganized, moving abruptly between concepts without a clear link. To improve, the authors should explain TPB in detail, justify its relevance to crop-pollination management, and integrate economic incentives with supporting data.

Suggested issue 7:

The manuscript is littered with many sentences which require rewording to bring clarity. I have listed some below and follow up is required on others:

Line 169: The sentence “Contribution refers to farmers’ observation the frequency of occurrence about pollinator’s visits to crops” is not well-worded and needs clarification for proper grammatical structure and meaning. Here's a suggested revision: “Contribution refers to farmers’ observations of the frequency of pollinator visits to crops”

Line 54: The sentence “Moreover, pollinator-dependent crops show higher agricultural expansion rates than other crops” is not well-worded and needs clarification for proper grammatical structure and meaning. Here's a suggested revision: “Suggestion: Additionally, crops that depend on pollinators exhibit higher rates of agricultural expansion compared to non-pollinator-dependent crops”

Line 86: The phrase "it to a large extent has higher potential" is awkward and not grammatically sound. It needs rephrasing for clarity and flow

Line 183: The sentence “The more diversified are farmers’ adoption pollinator-supporting practices, the more options they must avoid adverse effects induced by inadequate contribution, optimizing discrepancies between farmers’ cognition of pollinator dependence and researchers” needs rewording. Suggested re-wording: “The more diversified farmers’ adoption of pollinator-supporting practices, the greater their options for avoiding adverse effects caused by insufficient pollinator contributions. This approach also helps to optimize the discrepancies between farmers’ perceptions of pollinator dependence and those of researchers”

Line 268: The sentence “Among these crops, sunflower and melon are more rely on pollination mediated by insects” needs rewording. Suggested rewording “ Among these crops, sunflower and melon rely more on pollination mediated by insects.”

Line 283: The sentence “The start and end dates of the recruitment period for this study are 01/07/2021 and 31/08/2021, respectively.” is clear but could be improved for readability and formality. Suggested revision: "The recruitment period for this study began on July 1, 2021, and ended on August 31, 2021."

Line 76-81 To avoid repetitive use of 'on the other hand in two sentences following each other immediately, consider this ["While differences in perceptions between farmers and researchers or policymakers highlight an understanding and communication gap (Batie et al., 2009), these disparities also reflect variations in knowledge and background among local farmers. This is because the benefits of pollinator-supporting practices on yield can vary depending on management factors, and farmer behavior—an essential component of agricultural practices—is directly influenced by their local knowledge and experience (Batie et al., 2019)]

Line 308 : The sentence "However, the measurement farmers’ CPSM, which should attempt to consider specific regions and various crop types, is very difficult" is not clear. The sentence structure could be simplified for better readability. A more polished revision would be: "However, measuring farmers' CPSM, which should account for specific regions and various crop types, is challenging."

Line 663: The sentence “The enforcement agencies of pollination compensation will be refined to ensure the strict enforcement of pollination protection laws, and the central legislation 664 will be coordinated with the implementation of local departments” needs refinement. Suggested revision "The enforcement agencies responsible for pollination protection should be strengthened to ensure strict adherence to pollination protection laws, and coordination between national legislation and local departments should be improved to support effective implementation."

Line 583: [Farmers more affected by SN are less sensitive to the value of CPSM 582 in regulating crop yield and quality, therefore, SN does not support CPSM for farmers in Dengkou County (Hipólito et al., 2018).] Suggestion revision "Farmers who are more affected by SN are less sensitive to the value of CPSM in regulating crop yield and quality. Therefore, SN does not support CPSM for farmers in Dengkou County (Hipólito et al., 2018)."

Suggested issue 8: Line 269: In the methodology,

The sentence "unwritten regulations" may cause confusion, as it suggests rules that are not formally documented, which can be ambiguous in academic writing. To clarify, it’s better to use "informal regulations" if referring to practices that are followed but not codified. Additionally, the sentence uses "enough" twice, which makes it sound repetitive. To improve readability and flow, varying the language would help. The contraction "don’t" should also be avoided in formal writing, replaced with "do not" for a more professional tone. Lastly, the sentence structure could be refined to ensure greater clarity and formality. A revised version would be: "However, policies and informal regulations in Dengkou County do not give sufficient attention to crop-pollination services, and farmers lack adequate awareness of CPSM.

Suggested issue 9:

Line 290: The passage provides important ethical information but lacks clarity and coherence. The phrase "this study did not involve human or animal experimentation" is unnecessary since it contradicts earlier mentions of human participants. It would also benefit from clearer explanation about how informed consent was obtained and how respondents' data was protected. Suggested revision: "The Ethics Committee of the School of Economics and Management, Inner Mongolia Normal University, Hohhot, China, approved this study. All procedures involving human participants adhered to ethical standards, including the Declaration of Helsinki. Informed consent was obtained from all participants, and data were collected securely via online software to protect personal information."

Suggested issue 10: The phrase "267 questionnaires are returned" should be rephrased to improve readability, and the connection between the questionnaires and other data sources should be more clearly established. Suggested revision: "A total of 267 completed questionnaires were returned, all of which were deemed valid. The questionnaire covered elements related to CPSM and TPB, with specific questions listed in the Appendix (Table 1: Weights of the principles and chosen elements used in ICSM; Table 2: Descriptive statistics of the elements used to measure TPB constructs).

Suggested issue 11:

Placement of dollar sign not consistent. The dollar sign in the sentence is not correctly placed. It should come before the number for consistency with standard formatting. Example below [Pollination compensation standard in Dengkou County should be 152.27$, which is the lowest EI ($1900.27) of the optimal CPSM minus the difference between farmers’ income and expenditure.]

Suggested issue 12:

Check to see if the following were meant to be heading or sentences

• Motivate CPSM by improving farmers’ AT about pollinator

• Innovate and popularize technology about CPSM

**Do you want your identity to be public for this peer review?** For information about this choice, including consent withdrawal, please see our Privacy Policy

Reviewer #1: No

Reviewer #2: **Yes: ** Huang Jiaxing

Reviewer #3: **Yes: ** Richard Gyamfi Boakye

---

## [Decision Letter · Decision Letter 1]

Configuration analysis of crop-pollination service management: a novel insight from the theory of planned behavior

PONE-D-24-49557R1

Dear Dr. Chen,

We’re pleased to inform you that your manuscript has been judged scientifically suitable for publication and will be formally accepted for publication once it meets all outstanding technical requirements.

Kind regards,

Mehdi Rahimi, Ph.D.

Academic Editor

PLOS ONE

Additional Editor Comments (optional):

no changes are required from the authors in response to reviewer comments.

Reviewers' comments:

Reviewer's Responses to Questions

**Comments to the Author**

Reviewer #1: (No Response)

Reviewer #2: All comments have been addressed

Reviewer #3: All comments have been addressed

2. Is the manuscript technically sound, and do the data support the conclusions?

Reviewer #1: No

Reviewer #2: Yes

Reviewer #3: Yes

3. Has the statistical analysis been performed appropriately and rigorously?

Reviewer #1: No

Reviewer #2: Yes

Reviewer #3: Yes

4. Have the authors made all data underlying the findings in their manuscript fully available?

Reviewer #1: No

Reviewer #2: Yes

Reviewer #3: Yes

5. Is the manuscript presented in an intelligible fashion and written in standard English?

Reviewer #1: No

Reviewer #2: Yes

Reviewer #3: Yes

Reviewer #1: Although the author has made revisions and additions to the manuscript, simply observing and describing the relevant research content cannot directly support the research conclusions.

The importance of ecological factors such as pollinator diversity, environmental changes and large-scale agricultural changes is relatively limited in the behavioural research of crop pollination service management (CPSM), which is completely detached from the background of natural ecosystem pollination. The author focuses on farmers' views and behaviours towards crop pollination service management (CPSM) and socio-economic determinants, such as identifying feasible strategies to improve CPSM at the farmer level based on the theory of Theoretical Planning Behaviour (TPB). The aim is to explore how factors such as farmers' knowledge, attitudes, economic incentives and farm size affect their CPSM practices.

Therefore, the content and results of this study can contribute to sociology and management-related sciences and is more suitable for publication in social management and theoretical science journals. I strongly recommend the author to switch to professional journals related to social management and theoretical science.

Reviewer #2: Based on the revised version, the quality of the manuscript has improved. I only have a few further comments:

L74-77: It would be better to divide “artificial insect pollination” to “managed insect pollination” and “artificial pollination”, as they are two different pollination approaches.

L426: “FsQCA3.0 and R4.1.1 software is used”. Please add the reference for the two kinds of software used, for example, R4.1.1 (R Core Team, 20??). This information can be found in the software.

Appendix: A spelling error, change “scorces” to “scores”, check throughout the text.

Reviewer #3: Review Report for Manuscript: PONE-D-24-49557_R1

(By Dr Richard Gyamfi Boakye)

This report evaluates the revisions made by the authors in response to the previous recommendations for the manuscript titled ‘Configuration analysis of crop-pollination service management: A novel insight from the theory of planned behavior'. The recommendations made in the earlier manuscript are assessed to determine if they were properly addressed in the revised manuscript.

1. Cultural Factors

• Previous Recommendation: The manuscript should explore cultural influences on farmers' decision-making processes, including beliefs, traditions, and community norms, particularly in rural contexts like Dengkou County.

• Revised Response: The authors acknowledged the importance of cultural factors and stated that while subjective norms in the Theory of Planned Behavior (TPB) reflect social and cultural influences, the study did not delve deeply into cultural background due to funding limitations. The authors have committed to incorporating these cultural factors in future studies.

• Assessment: The suggestion has not been fully implemented, but the authors have recognized its significance and proposed addressing it in future research. This acknowledgement is deemed satisfactory.

2. Ecological Data

• Previous Recommendation: The authors were advised to integrate ecological data, such as pollinator diversity and abundance, to correlate management practices with actual pollination outcomes.

• Revised Response: The authors explained that the study focused on socio-economic and behavioral factors, aligning with the scope of their research objectives. They acknowledged that ecological data would be a valuable addition in future research.

• Assessment: The recommendation was acknowledged, but the suggestion was not fully implemented due to study constraints. I acknowledge that and express satisfaction.

3. Longitudinal Approach

• Previous Recommendation: A longitudinal study could offer deeper insights into changes in farmers' behavior over time and the long-term impacts of interventions.

• Revised Response: The authors maintained that their study used a cross-sectional design, which was appropriate for their research objectives. They expressed intent to explore a longitudinal approach in future research.

• Assessment: The authors justified the exclusion of the longitudinal approach and did not implement it in the revised manuscript. Their justification is satisfactory.

4. Consultation Process with Experts

• Previous Recommendation: The consultation with regional pollinator experts and non-academic stakeholders should be explicitly described in the methodology section for clarity and transparency.

• Revised Response: The authors agreed with this suggestion and moved the description of expert consultations from the supplementary materials to the methodology section. They also clarified how these consultations contributed to the development of the study's framework.

• Assessment: This recommendation has been fully addressed, and the consultation process is now clearly stated in the methodology section.

5. Equations and Methodology

• Previous Recommendation: The detailed equations used in the study should be simplified in the main text, with supporting materials placed in the supplementary section.

• Revised Response: The authors followed this recommendation by summarizing the methodology in plain language in the main text and placing the detailed equations in the supplementary material.

• Assessment: The authors have implemented this recommendation effectively.

6. Theory of Planned Behavior (TPB)

• Previous Recommendation: A more thorough introduction of the TPB and its extension with economic incentives should be provided for readers unfamiliar with the theory.

• Revised Response: The authors expanded their explanation of the TPB, including more context to make the theory accessible to a broader audience. They also clarified the extension of TPB using economic incentives.

• Assessment: The authors have addressed this recommendation in the revised manuscript.

7. Suggested revision of sentences

Generally, all sentenced needing revision have been revised accordingly

Overall Recommendation for Publication

Based on the revisions made by the authors, most of the suggestions were considered, especially the inclusion of expert consultations in methodology and the restructuring of the equations. However, some points, such as incorporating cultural influences and ecological data, were either beyond the scope of the current study or planned for future research. Given these revisions, I recommend acceptance for publication. The paper demonstrates significant improvements, particularly in methodological transparency, but could benefit from the integration of cultural and ecological factors in future work.

**Do you want your identity to be public for this peer review?** For information about this choice, including consent withdrawal, please see our Privacy Policy

Reviewer #1: No

Reviewer #2: No

Reviewer #3: **Yes: ** Richard Gyamfi Boakye

---

## [Editor Report · Acceptance letter]

PONE-D-24-49557R1

PLOS ONE

Dear Dr. Chen,

I'm pleased to inform you that your manuscript has been deemed suitable for publication in PLOS ONE. Congratulations! Your manuscript is now being handed over to our production team.

Kind regards,

on behalf of

Associate Prof. Mehdi Rahimi

Academic Editor

PLOS ONE